# The health effects associated with physical, sexual and psychological gender-based violence against men and women: a Burden of Proof study

The health impacts of exposure to physical, sexual or psychological gender-based violence (GBV) against men and women are substantial yet not well delineated. We systematically reviewed and meta-analysed 40 studies to evaluate the associations between GBV (including but not limited to intimate partner violence) and eight health outcomes: sexually transmitted infections excluding HIV, maternal abortion and miscarriage, HIV/AIDS, major depressive disorder, anxiety disorders, drug use disorders, alcohol use disorders and self-harm. Using the Burden of Proof methods, we generated conservative metrics of association—including star ratings from one to five—reflecting both effect size and evidence strength. Sexual violence was associated with six outcomes, with moderate, three-star evidence of association for sexually transmitted infections, maternal abortion and miscarriage, and major depressive disorder—increasing the risk by at least 104%, 101% and 50%, respectively. Seven outcomes were associated with physical GBV with two- or one-star associations, reflecting weak associations and/or inconsistent evidence. Of the four health outcomes analysed in relation to psychological GBV, one, major depressive disorder, had a significant association with a one-star rating. These findings emphasize the serious health consequences of GBV for survivors and the necessity of additional data to further our understanding of this complex public health issue.

Gender-based violence (GBV) is a human rights abuse and public health issue associated with substantial morbidity and mortality globally[1,2]. GBV is defined as violence directed at an individual on the basis of their biological sex, gender identity, gender expression or failure to adhere to socially defined norms of masculinity and femininity[3]. It can take a variety of forms including physical, sexual and psychological violence and is perpetrated by a diverse array of actors ranging from intimate partners and trusted individuals to complete strangers. Exposure to

GBV is associated with harmful health outcomes that can manifest at multiple stages and in multiple forms over the life of survivors. Health impacts can be severe and, in some cases, irreversible, requiring life-long medical treatment to alleviate suffering. Survivors are too often not able to achieve their full potential and dedicate large portions of their lives and resources to maintaining their healing process[1,2,4]. The adverse impacts extend beyond the individuals immediately harmed, as families and wider communities suffer from detrimental

✉e-mail: gakidou@uw.edu

secondary consequences and long-term intergenerational impacts of GBV[2,5,6].

Over the past 50 years, research on the health effects of GBV has expanded dramatically, with numerous studies recognizing GBV as a risk factor for a range of conditions including depression, alcohol use disorders and post-traumatic stress disorders[4,7–10]. Despite this progress, substantial research gaps remain that hinder our ability to more effectively address the impacts of GBV. For example, not all existing evidence properly controls for the timing between exposure and outcome, with few longitudinal studies being available, limiting the evidence base for establishing causality between GBV and health outcomes. Moreover, most existing studies focus on intimate partner violence (IPV) against women, and most often do not distinguish the health consequences associated with specific types of violence (for example, the consequences of sexual, physical or psychological violence separately). Additionally, while the role of men as common perpetrators of GBV has long been recognized, their position as victims has historically been understudied, resulting in limited data availability on the health consequences for male survivors[11]. These gaps showcase a clear need to comprehensively examine the current landscape of peer-reviewed literature around the health impacts of GBV in order to consolidate and analyse the available evidence beyond the field's long-standing focus on IPV against women.

Our study—which contributes to the goals of the Lancet Commission on Gender-Based Violence and Maltreatment of Young People[2] to expand the consideration of GBV as an important global health concern for all people—provides a systematic review of the health effects associated with exposure to physical, sexual and psychological GBV against men and women. This work builds off a prior effort to quantify and evaluate the health effects of IPV and childhood sexual abuse[12,13] by taking a broader lens on GBV exposure and its associated health consequences. We extended our analysis of GBV exposure to include physical, sexual and psychological GBV against women, regardless of the perpetrator, and to include data on men's exposure to physical and psychological IPV and sexual violence. Expanding our exposure scope allows us to disentangle the different manifestations of GBV to provide a more accurate representation of the complex relationships between GBV exposure and health outcomes beyond the common focus of IPV among women and, in the future, to allow for a more comprehensive estimation of the disease burden attributable to GBV.

Drawing from a systematic review of the literature published between 1 January 1970 and 31 January 2024, we estimated the associations between physical, sexual and psychological GBV and a range of health outcomes using the Burden of Proof methodology developed by Zheng et al.[14]. This methodology allows us to both systematically evaluate the potential association between the exposure of interest and a given health outcome and quantify the strength of the underlying evidence. In addition to producing conventional measures of association, the Burden of Proof methodology generates conservative measures that account for both known and unknown sources of heterogeneity across input studies. The Burden of Proof Risk Function (BPRF) can be translated into both a Risk–Outcome Score (ROS) and an estimate of the minimum percentage of increased health risk attributable to GBV exposure. The ROS communicates both the magnitude of the association and the strength of the underlying data, with greater positive values reflecting a larger effect size and/or more consistent evidence. The ROS can in turn be converted into easily comparable and interpretable star ratings ranging from one (weak) to five (strong) that categorize significant associations according to effect size and evidence strength.

The risk–outcome associations presented here contribute to more adequately capturing the health burden associated with various forms of GBV. Our findings highlight the long-lasting consequences of GBV, the need for accelerated violence prevention policies and programmes,

and priority areas for making health resources and interventions available to survivors. The main findings and policy implications of our study are summarized in Table 1.

## Results

### Overview

We screened 75,331 studies in a systematic review of the literature available across seven databases, aimed at assessing the health impacts of any form of violence over the life-course. Across these studies, we identified enough data to feasibly analyse a total of eight health outcomes in relation to physical GBV, sexual violence and/or psychological GBV (that is, risk–outcome associations with at least three studies available, following the Burden of Proof methodology)[14]. These health outcomes include mental disorders (that is, anxiety disorders and major depressive disorder), substance use disorders (that is, alcohol use disorders and drug use disorders), maternal and reproductive outcomes (that is, maternal abortion and miscarriage, HIV/AIDS, and other sexually transmitted infections (STIs)), and self-harm. In the present analyses of these eight health outcomes, we have meta-analysed 40 unique studies, of which 26 measured physical GBV, 25 sexual violence and 14 psychological GBV. The risk–outcome pair most frequently investigated was sexual violence and HIV/AIDS ($n = 10$), followed by physical GBV and maternal abortion and miscarriage ($n = 8$). Further details and inclusion and exclusion criteria are presented in a PRISMA diagram (Fig. 1); in brief, included studies needed to employ a study design that allowed the research team to establish temporality and to meet requirements for study sample composition and exposure/outcome definitions. The characteristics of all included studies can be found in Supplementary Table 1.

Across all 103 observations, 82 pertained to women, 18 to men, and three to women and men combined. The effect sizes are delineated according to instances of GBV that occurred ever, in the past three years, in the past year, in the past six months, in the past three months, and during pregnancy in a few instances. The majority of observations ($n = 68$) reported partner- or former-partner-perpetrated GBV, followed by GBV from unspecified or unrestricted perpetrators ($n = 33$).

In addition to the main models' results presented in Table 2 and Figs. 2–4, we ran sensitivity analyses for all exposures and health outcomes considering the type of perpetrator, the gender of participants, variations in outcomes, and excluding pregnancy recall when we had enough studies to do so (Extended Data Figs. 1–3, Supplementary Tables 2–8 and Supplementary Figs. 1–21).

### Physical GBV

A total of eight eligible outcomes were identified with enough data to evaluate their relationship to physical GBV (Table 2 and Fig. 2). Importantly, due to the difficulty in identifying gendered motivations of physical violence against men, the included male-specific observations are narrowly focused on intimate partner physical violence against men to reflect this study's emphasis on gender-based physical violence. The associations between exposure to physical GBV and drug use disorders and HIV/AIDS received two-star ratings, suggesting moderately weak evidence of an association (Table 2). For drug use disorders, we identified eight relevant observations from six cohort studies[15–20] with women ($n = 6$) and men ($n = 2$) participants, including seven observations that measured physical IPV. Our conservative interpretation of the evidence for this two-star relationship suggests at least a 20% increase in risk of drug use disorders (BPRF = 1.20; ROS = 0.9) given exposure to physical GBV. For HIV/AIDS, all four studies identified focused on physical IPV against women[21–24]. Within the context of these restricted data, we found that physical GBV increased HIV/AIDS risk by at least 15% (BPRF = 1.15; ROS = 0.07) (Table 2 and Supplementary Table 4).

We identified a further five health outcomes with evidence of a significant association with physical GBV exposure: alcohol use disorders ($n = 3$)[15–17], major depressive disorder ($n = 4$)[17,25–27], STIs ($n = 3$)[28–30],

## Table 1 | Policy summary

| Background | Exposure to GBV is one of the world's most prevalent and pervasive human rights abuses, resulting in severe consequences for individuals' health and well-being. Yet the impacts of exposure to physical, sexual or psychological GBV are still not well understood. Here we draw upon a systematic review to evaluate the associations between GBV and health outcomes that have been described in peer-reviewed journals. In a Burden of Proof meta-analysis, we estimated the magnitude of the distinct associations between exposure to physical, sexual and psychological GBV—including but not limited to IPV—and eight health outcomes and evaluated the strength of the evidence underpinning these associations. |
|---|---|
| Main findings and limitations | We evaluated the associations between physical, sexual or psychological GBV and six, eight and four health outcomes, respectively. Each of the risk–outcome pairs we investigated was supported by at least three studies. The associations between sexual violence and STIs excluding HIV/AIDS, maternal abortion and miscarriage, and major depressive disorder received moderate (three-star) ratings. Exposure to this type of violence was found to increase the risk for each of these conditions by at least 104%, 100% and 50%, respectively. Significant associations were also found between sexual violence and HIV/AIDS, anxiety disorders and drug use disorders, but the evidence was rated as weak (one-star rating). Associations between physical GBV and drug use disorders and HIV/AIDS received two-star ratings, classified as moderately weak; and one-star ratings were assessed for the relationship of physical GBV to five other outcomes (alcohol use disorders, major depressive disorder, anxiety disorders, STIs, and maternal abortion and miscarriage). Exposure to psychological GBV was associated with major depressive disorder (one-star rating), but no evidence of association was found with self-harm, maternal abortion and miscarriage, or drug use disorders. Similarly, insufficient evidence of a significant association was found between physical GBV and self-harm.<br><br>The primary limitations of this study are related to model parameters and data availability. Definitions of violence exposure differed between studies, mainly in relation to recall timing and survey methodologies used to evaluate exposure. To the extent possible, bias covariates were created and included in the model to account for study-level differences. Physical, sexual and psychological GBV exposure was evaluated as a dichotomous risk, which did not allow us to consider differences in frequency and severity. It is important that future research includes dose–response analyses for a better understanding of the health effects of exposure to GBV. Also, despite a literature search and the use of the most recently available data, this study does not capture the full breadth of the health burden associated with violence since several health outcomes identified were not included in our analysis due to insufficient data. |
| Policy implications | This review emphasizes the long-lasting negative health consequences of experiencing GBV and the importance of timely, comprehensive data on associations between GBV and health outcomes, continuing to expand the landscape of evidence on GBV's health consequences beyond IPV against women. In particular, this analysis showed that the existing data landscape systematically overlooks GBV victimization as a health risk factor for men. While women are disproportionately exposed to GBV compared with men and, as such, experience a correspondingly disproportionate proportion of GBV's health toll, GBV perpetrated against men must not be overlooked in future research, policy or programming. The analysis also showed that there is a clear paucity of data on the health effects of psychological GBV compared with data on physical or sexual GBV, leaving a substantial need for strategies to better detect, address and understand psychological GBV as a policy and research priority. It is essential to amplify actions and commitment from global to regional, national and community-based strategies for prevention, recovery and justice for individuals, groups or couples, also interrupting the intergenerational cycles of violence. Investing in multi-sectoral interventions, such as improved GBV screening and referral to trauma-informed care, can make an important difference by prioritizing safety, patient autonomy, shared decision-making and empowerment. |

anxiety disorders ($n = 3$)[17,25,27], and maternal abortion and miscarriage ($n = 8$)[31–38]. However, the ROSs for these relationships ranged from −0.002 (for alcohol use disorders) to −0.23 (for maternal abortion and miscarriage), rendering them one-star risk–outcome pairs, indicative of weak associations and/or a lack of consistent evidence (Table 2). The three eligible studies with data on STIs and physical GBV all reported only women-specific data. In comparison, when we excluded data on men, focusing instead on women-specific data in sensitivity analyses for major depressive disorder and anxiety disorders, we found slightly stronger evidence of associations, with two-star ratings (ROS = 0.01 and 0.02, respectively) (Extended Data Fig. 1, Supplementary Tables 5 and 6 and Supplementary Fig. 2). While we did not have enough men-specific data for anxiety disorders to run a separate sensitivity analysis just for men, there was insufficient evidence of a statistically significant association between physical GBV and major depressive disorder when restricting the data only to men (Extended Data Fig. 1, Supplementary Table 5 and Supplementary Fig. 1). This distinction was also highlighted in our primary model for major depressive disorder, in which the inclusion of men in the analytical sample was flagged and adjusted for as a significant source of bias (Table 2). For maternal abortion and miscarriage, crude effect sizes[32,34,36] and effect sizes that focused on induced abortion[31,32,36] were flagged as significant bias covariates and adjusted for in the final models (Table 2). However, our one-star association was consistent across all the sensitivity analyses for this outcome, including the model with data restricted to only effect sizes for induced abortion. The pooled relative risk (RR) was also largely consistent across sensitivity analyses, only dipping below 2.00 when we restricted the data to exposures of physical GBV outside of pregnancy (that is, no

pregnancy recall) (Extended Data Fig. 1, Supplementary Table 7 and Supplementary Fig. 3).

Lastly, the association between exposure to physical GBV and self-harm, informed by three eligible studies[29,39,40], was found to have insufficient evidence to support an association. The conventional 95% uncertainty intervals (UIs) included an RR of 1 (that is, the null), reflecting no credible evidence of a relationship (Table 2 and Fig. 2).

### Sexual violence

Out of the six health outcomes evaluated, all were found to be associated with exposure to sexual violence (Table 2 and Fig. 3). Among these, three presented moderate (three stars) evidence of a relationship with sexual violence: STIs, maternal abortion and miscarriage, and major depressive disorder (Table 2). The association between sexual violence and STIs, the strongest risk–outcome relationship (ROS = 0.4), was informed by four observations from four prospective cohort studies[28–30,41] involving only women participants, while the association between sexual violence and major depressive disorder was informed by five observations from one case–control study and four prospective cohort studies spanning both women and men[27,42–44]. The BPRF metrics suggest that the risk of STIs and major depressive disorder increased by a minimum of 104% and 50%, respectively, among individuals exposed to sexual violence (BPRF = 2.04 and 1.50; Table 2 and Fig. 3).

Similarly, our estimates suggest that sexual violence increased the risk of maternal abortion and miscarriage by at least 100% (BPRF = 2.00; Table 2 and Fig. 3). We identified six relevant observations from five studies[32,35,36,38,41] for maternal abortion and miscarriage, including three observations that focused on sexual IPV[35,36,38]. When limiting our

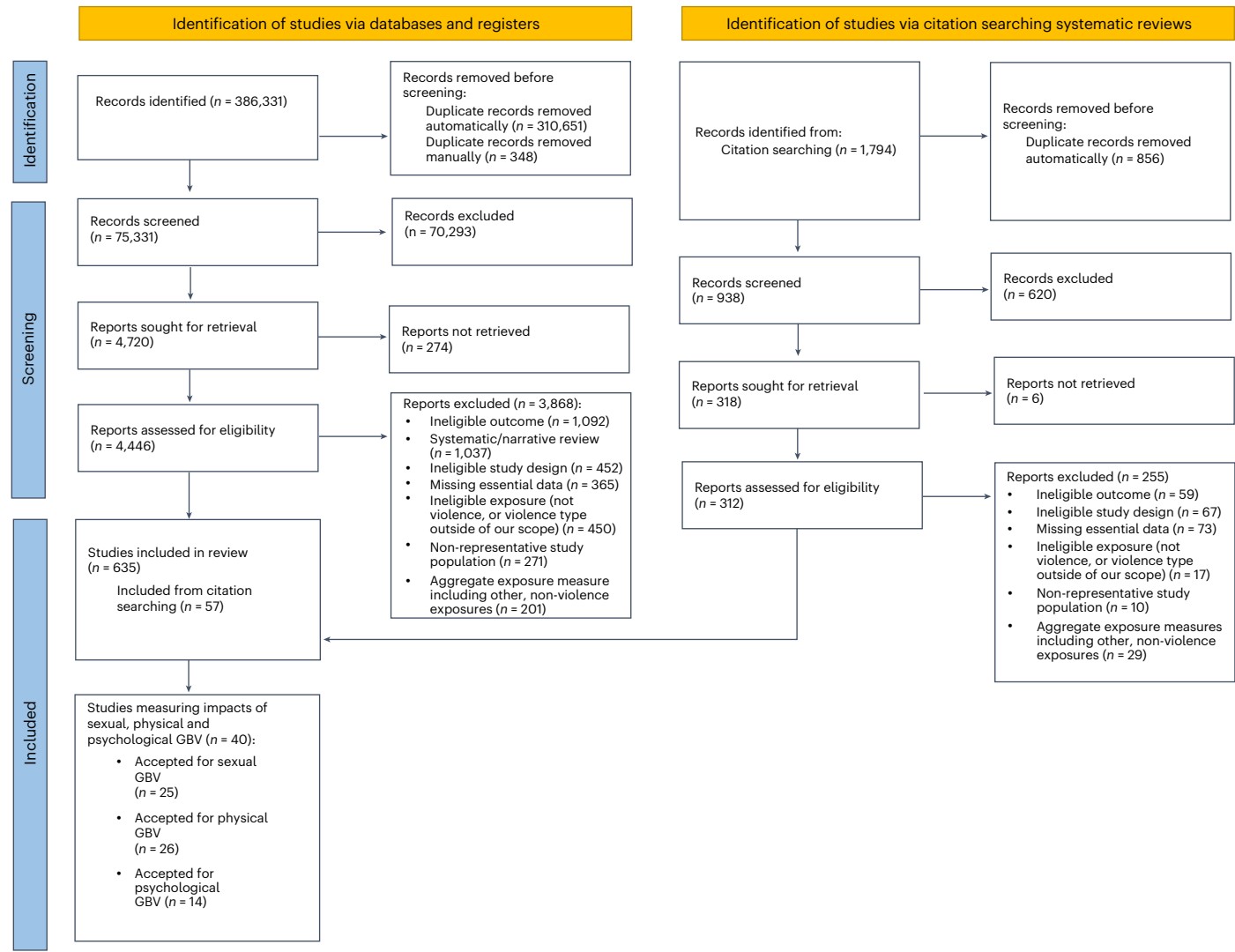

**Fig. 1 | Flow diagram of a systematic review on health effects associated with physical, sexual and psychological GBV against men and women.** The PRISMA flow diagram shows the data-seeking approach undertaken as a part of our review, which sought to identify all literature reporting on the health effects of GBV, violence against women, and violence against children and young people. Studies specifically measuring the health impacts of physical, sexual and psychological GBV against men and women were used for this analysis.

analysis to only the IPV-related data points, we found slightly lower strength of evidence at two stars (Extended Data Fig. 2, Supplementary Table 7 and Supplementary Fig. 9). A sensitivity analysis restricted to three studies with data on induced abortion resulted in a four-star rating (ROS = 0.41), while one limited to cases of miscarriage resulted in a two-star rating (ROS = 0.04; Extended Data Fig. 2, Supplementary Table 7 and Supplementary Fig. 10).

In contrast, the associations between sexual violence and drug use disorders, anxiety and HIV/AIDS were each rated one star, suggesting weak and/or inconsistent evidence of a relationship (Table 2 and Fig. 3). The relationship between HIV/AIDS and sexual violence was the most widely studied association of all those included in the present analysis, with 11 observations from one case–cohort, four cohort and five case–control studies[21,23,24,41,45–50]. Most of these studies had been conducted in African countries and focused solely on women, while three narrowed in on partner-perpetrated sexual violence. A sexual-IPV-specific sensitivity analysis resulted in a star rating of two (ROS = 0.05) and a minimum increased risk of 10% (BPRF = 1.1) (Extended Data Fig. 2, Supplementary Table 4 and Supplementary Fig. 9). We included six observations from four prospective cohorts[16,18,20,43] for drug use disorders and four observations from one case–control and two cohort studies[27,43,51]

for anxiety disorders. The input data were heavily women-specific and, with two exceptions, did not restrict their exposure definitions on the basis of the identity of the perpetrators. Our findings were consistent across sensitivity analyses, including when restricting to only observations from women (Extended Data Fig. 2, Supplementary Tables 3 and 6 and Supplementary Fig. 11).

**Psychological GBV**

Across the forms of GBV investigated, we identified the fewest overall number of studies for psychological forms of GBV, which resulted in only four health outcomes with enough data to analyse (Table 2 and Fig. 4). Akin to our review of physical GBV, the landscape of psychological GBV data among men is hindered by a lack of distinction between psychological violence with gender-related motivations and other forms of psychological violence. Our analysis of psychological GBV among male survivors therefore concentrated on psychological IPV, which is inherently gender-related, while the focus among female survivors was broader to align with the definitions of psychological GBV published in the existing literature. Within this context, only major depressive disorder was found to be associated with exposure to psychological GBV (Table 2). However, this statistically significant association

**Table 2 | Strength of the evidence for the relationship between GBV exposure and eight health outcomes**

| Health outcome | Mean RR | 95% UI for the mean RR without γ | 95% UI for the mean RR with γ | BPRF | ROS | Star rating | Pub. bias | No. of studies (observations) | Selected covariates |
|---|---|---|---|---|---|---|---|---|---|
| **Physical GBV** | | | | | | | | | |
| Drug use disorders | 1.84 | 1.50–2.25 | 1.11–3.06 | 1.20 | 0.9 | ☆☆ | No | 6 (8) | None |
| HIV/AIDS | 1.65 | 1.37–1.98 | 1.08–2.52 | 1.15 | 0.07 | ☆☆ | No | 4 (4) | None |
| Alcohol use disorders | 1.51 | 1.21–1.89 | 0.92–2.48 | 1.00 | −0.002 | ☆ | No | 3 (5) | None |
| Major depressive disorder | 1.53 | 1.21–1.94 | 0.83–2.82 | 0.92 | −0.04 | ☆ | No | 4 (7) | Men included in the effect size |
| STIs excluding HIV | 1.64 | 1.11–2.42 | 0.66–4.10 | 0.76 | −0.14 | ☆ | No | 3 (3) | None |
| Anxiety disorders | 1.82 | 1.19–2.77 | 0.60–5.51 | 0.72 | −0.17 | ☆ | No | 3 (7) | Outcome is defined as PTSD |
| Maternal abortion and miscarriage | 2.76 | 1.74–4.38 | 0.48–15.99 | 0.63 | −0.23 | ☆ | No | 8 (8) | Unadjusted effect size; outcome is defined as induced abortion |
| Self-harm | 3.17 | 0.99–10.12 | 0.13–79.57 | 0.21 | N/A | | No | 3 (3) | None |
| **Sexual violence** | | | | | | | | | |
| STIs excluding HIV | 2.95 | 2.39–3.65 | 1.90–4.58 | 2.04 | 0.36 | ☆☆☆ | No | 4 (4) | None |
| Maternal abortion and miscarriage | 2.53 | 2.20–2.91 | 1.91–3.35 | 2 | 0.35 | ☆☆☆ | No | 5 (6) | None |
| Major depressive disorder | 1.99 | 1.71–2.31 | 1.43–2.77 | 1.5 | 0.2 | ☆☆☆ | No | 4 (5) | None |
| HIV/AIDS | 1.85 | 1.42–2.39 | 0.73–4.68 | 0.85 | −0.08 | ☆ | No | 10 (11) | Unadjusted for age |
| Anxiety disorders | 2.86 | 1.66–4.92 | 0.61–13.34 | 0.78 | −0.12 | ☆ | No | 3 (4) | None |
| Drug use disorders | 1.69 | 1.22–2.36 | 0.66–4.33 | 0.77 | −0.13 | ☆ | No | 4 (6) | None |
| **Psychological GBV** | | | | | | | | | |
| Major depressive disorder | 1.66 | 1.06–2.60 | 0.46–5.95 | 0.57 | −0.28 | ☆ | No | 3 (5) | None |
| Maternal abortion and miscarriage | 1.34 | 0.91–1.96 | 0.38–4.70 | 0.47 | N/A | | No | 6 (7) | Unadjusted for age; unadjusted effect size |
| Drug use disorders | 1.49 | 0.96–2.30 | 0.43–5.09 | 0.53 | N/A | | No | 3 (6) | None |
| Self-harm | 1.56 | 0.94–2.61 | 0.48–5.07 | 0.58 | N/A | | No | 3 (4) | None |

The reported mean RR reflects the risk an individual who has experienced sexual, physical or psychological GBV has of developing the health outcome in the corresponding row relative to that of someone who has not been exposed to the given form of violence. The 95% UI for the mean RR without γ refers to the 95% UI that is estimated without fully incorporating between-study heterogeneity, while the 95% UI for the mean RR with γ refers to the 95% UI that is estimated fully incorporating between-study heterogeneity and the uncertainty around quantified between-study heterogeneity. The BPRF is calculated for risk–outcome pairs that were found to be significant when estimating a conventional RR and 95% UI (95% UI without γ). As the conservative estimate of risk given existing evidence, it corresponds to the fifth-quantile RR estimate incorporating between-study heterogeneity closest to the null value of 1. The percentage of excess risk is derived as (BPRF −1)×100. The ROS is derived as the signed ln(BPRF)/2. It reflects the strength of the available evidence and is translated into a star rating ranging from 1 (weak evidence) to 5 (strong and consistent evidence) on the basis of predetermined thresholds documented elsewhere. Risk–outcome pairs that do not meet the standard for calculating the BPRF are not assigned a star rating. The risk of publication bias is flagged on the basis of the results of Egger's regression and should inform the interpretation of the results. The selected bias covariates were flagged as reflecting significant sources of systematic bias and were adjusted for in the final models. In bold are the GBV exposure types.

was informed by three cohort studies[25,26,52] with weak underlying evidence as a one-star association (BPRF = 0.57; ROS = −0.28). For the other outcomes with enough data to analyse—self-harm (n = 3)[29,40,53], maternal abortion and miscarriage (n = 6)[32,35–38,54], and drug use disorders (n = 3)[15,18,53]—no credible evidence of an association was found with psychological GBV on the basis of conventional estimates of RR uncertainty.

Due to the limited availability of data, few sensitivity analyses were feasible. However, for maternal abortion and miscarriage, when we limited the input data to only studies examining induced abortion, our analysis suggested a significant relationship between this outcome and psychological GBV with a one-star association (ROS = −0.04) (Extended Data Fig. 3, Supplementary Table 7 and Supplementary Fig. 16).

**Comparison across forms of GBV**
Among the GBV exposure types investigated here, we identified the greatest number of health outcomes meeting the minimum

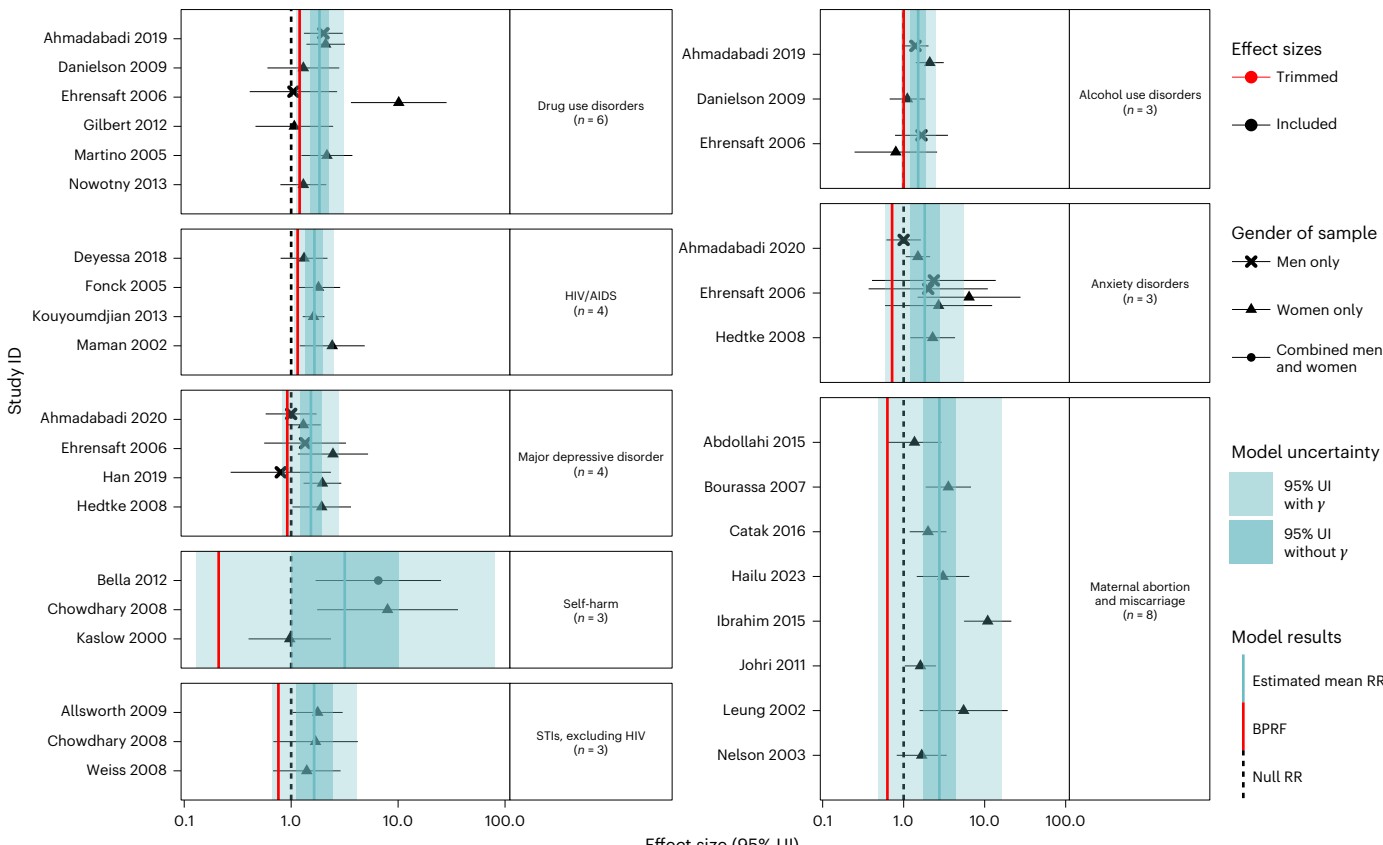

**Fig. 2 | Forest plots for physical GBV and health outcomes identified through a systematic review of the literature.** The shape of each point indicates the gender of the sample (women only, men only, and combined women and men). Light blue shading corresponds to the 95% UI incorporating between-study heterogeneity; dark blue shading corresponds to the 95% UI without between-study heterogeneity. The black vertical dashed lines reflect the null RR value (one), and the red vertical lines show the BPRF at the fifth quantile for these harmful risk–outcome associations. The black data points and horizontal lines each correspond to an effect size and 95% UI from the included study identified on the y axis. We included multiple observations from a single study when effects were reported by gender and health outcome (anxiety disorders and PTSD). Supplementary Table 1 contains more details on the observations included from each study.

requirement of three studies to analyse in association with physical GBV (Fig. 5). However, the risk–outcome pairs with the strongest evidence of associations were all related to sexual violence. Among the outcomes identified, maternal abortion and miscarriage, major depressive disorder, and drug use disorders had enough data to study across all three forms of GBV (Fig. 5). Of these three, our analysis identified only major depressive disorder as associated with all three exposures, with the evidence of association rated as weak (one star) for physical GBV and psychological GBV (ROS = −0.04 and −0.28, respectively) and moderate (three stars, ROS = 0.2) for sexual violence (Table 2 and Fig. 5). The analysis found no credible evidence of an association between psychological GBV and both maternal abortion and miscarriage and drug use disorder. Furthermore, it was feasible to analyse STIs, HIV/AIDS and anxiety disorders only in relation to physical and sexual GBV. Likewise, self-harm had enough data to analyse only in relation to psychological and physical GBV—and showed no credible evidence of an association with either—while alcohol use disorders had enough data only in relation to physical GBV.

## Discussion

In this systematic review and meta-analysis, we assessed the evidence base on the health impacts of exposure to physical, sexual and psychological GBV. The analysis draws upon peer-reviewed literature indexed across seven databases and published over the course of 54 years to evaluate the associations between three sub-types of GBV and eight health outcomes in addition to the strength of the underlying

evidence. The eight health outcomes investigated reflect the relationships that have been examined in peer-reviewed literature to date, encompassing all those with relevant data from a minimum of three eligible studies. We identified an association between physical GBV and seven health outcomes—drug use disorders, HIV/AIDS, alcohol use disorders, major depressive disorder, STIs, anxiety disorders, and abortion and miscarriage—although these associations and/or their underlying evidence were rated as weak, receiving one- and two-star ratings in the Burden of Proof framework. We also found evidence of an association between sexual violence and all aforementioned health outcomes except alcohol use disorders, which were not examined in relation to sexual violence due to a lack of applicable data. The associations between sexual violence and STIs, abortion and miscarriage, and major depressive disorder all received three-star ratings, reflecting a moderately strong effect size and/or underlying evidence; this was the highest star rating of any risk–outcome relationship identified in the present analysis. Conversely, of the four health outcomes feasible to analyse in relation to psychological GBV, only major depressive disorder was found to be significantly associated, albeit with weak (one star) evidence of an association.

Overall, our analysis confirmed past findings about the relationship between exposure to GBV and poor mental health[7–10] while highlighting the fact that the health effects of GBV extend beyond mental health, including substance abuse disorders and maternal and reproductive health. Overwhelmingly, however, the consistently weak evidence of associations between GBV and health outcomes

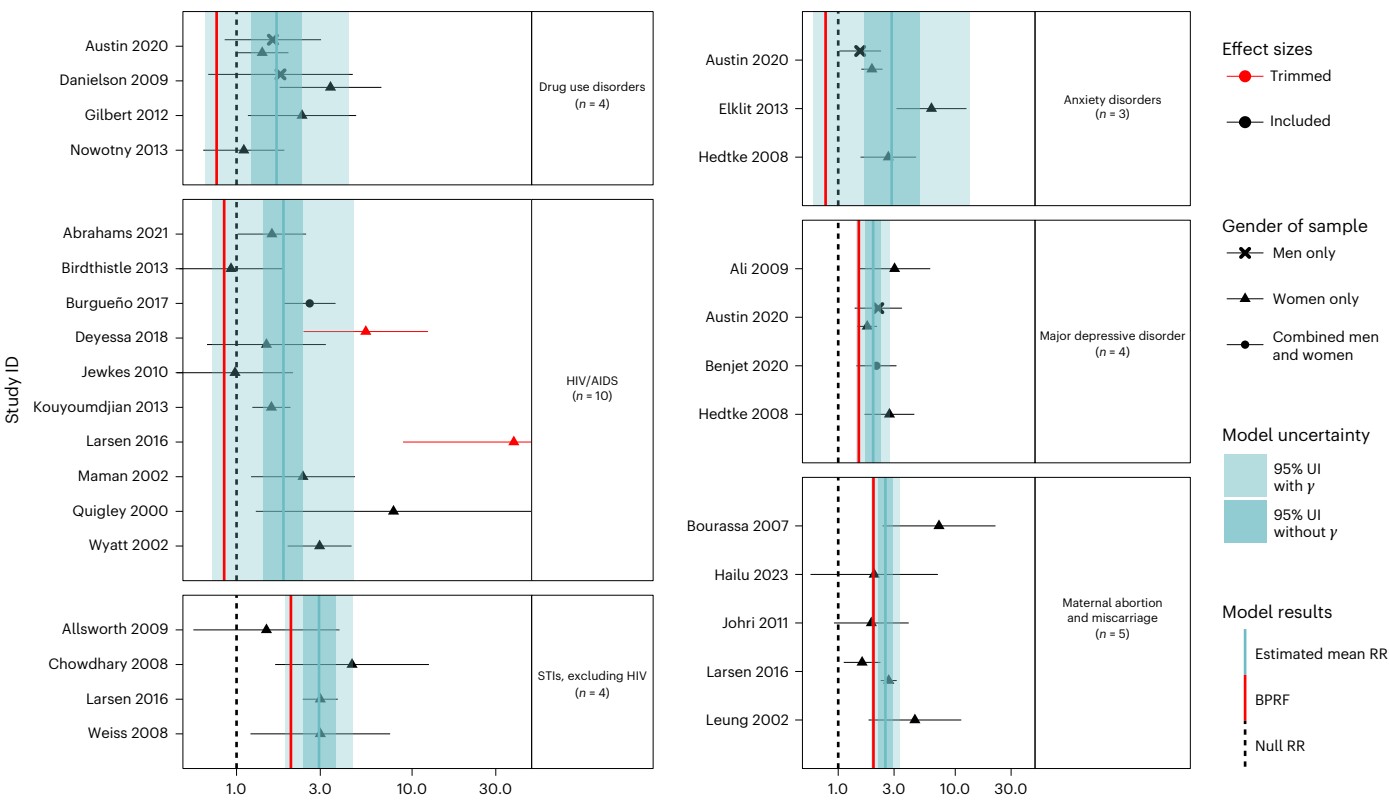

**Fig. 3 | Forest plots for sexual violence and health outcomes identified through a systematic review of the literature.** The shape of each point indicates the gender of the sample (women only, men only, and combined women and men). Light blue shading corresponds to the 95% UI incorporating between-study heterogeneity; dark blue shading corresponds to the 95% UI without between-study heterogeneity. The black vertical dashed lines reflect the null RR value (one), and the red vertical lines show the BPRF at the fifth quantile for these harmful risk–outcome associations. The black data points and horizontal lines each correspond to an effect size and 95% UI from the included study identified on the *y* axis. We included multiple observations from a single study when effects were reported by gender, health outcome (induced abortion and spontaneous abortion) and perpetrator. Supplementary Table 1 contains more details on the observations included from each study.

identified by our methods that incorporate between-study heterogeneity into the results illuminates the need for more research to further our understanding of known health associations to GBV and to extend the evidence on GBV's health consequences beyond IPV against women. In the studies we analysed, we did not restrict the inclusion of data points on the basis of the gender of participants but rather focused on addressing exposure to forms of violence that could be motivated by the survivor's gender. Our work thus stands out from previous meta-analyses[55] by incorporating effect sizes for GBV against men into our estimates of adverse health outcomes associated with physical, sexual and psychological GBV. Specifically, we included 11 data points examining physical IPV against men or against both men and women combined, four data points on psychological IPV against men, and six data points on sexual violence against men or against both men and women combined. Despite their inclusion, these data points are still minimal compared with the number of women-specific data points available, reflecting an imbalance in the existing data landscape that systematically overlooks GBV victimization as a health risk factor for men[56]. However, the consequences of GBV—regardless of the gender of the victim—are demonstrated by our present results, even in the context of our highly conservative interpretation of the evidence available. In lieu of a broader evidence base for men to allow for further granularity, we have to assume that there are no differences in the health consequences of GBV victimization between men and women, despite broad consensus on the gendered characteristics of the associated health outcomes[5,57,58] that make this assumption unlikely and deeply flawed. To begin understanding and responding to these possible

differences, GBV perpetrated against men must not be overlooked in future research, policy or programming.

In addition to the women-specific focus of existing literature, there is a clear lack of data on the health effects of psychological GBV compared with data on physical or sexual GBV. While any form of GBV is subject to under-reporting and under-detection[59], psychological GBV is particularly overlooked, with substantial inconsistency in the definition and recognition of this form of violence[60]. Additionally, while many of the studies identified in our present analysis use self-reported exposure to GBV to identify survivors, psychological violence manifests in ways that may be challenging for the survivors themselves to identify[61] and has been linked to feelings of shame and stigma[62–65]. Moreover, in addition to psychological GBV, we identified weak to moderate evidence of associations with sexual and physical GBV across all health outcomes analysed. These ratings can be attributed in part to both a high degree of variability in the underlying data and an overall low number of included studies. Out of 18 models across all three of our exposures of interest, only five were informed by data from more than five studies. Our inclusion criteria were established to identify high-quality studies with relevant exposure and outcome definitions, but relatively few studies—particularly when compared with other important health risk factors[66,67]—met our criteria. Even fewer studies met the gold standard for data on the health impacts of GBV: representative prospective cohort studies linked to multiple data systems. These studies require extended follow-up to monitor outcome development, sustained participant engagement, and safe and ethical data linkage, and they entail considerable financial costs[68], but they are

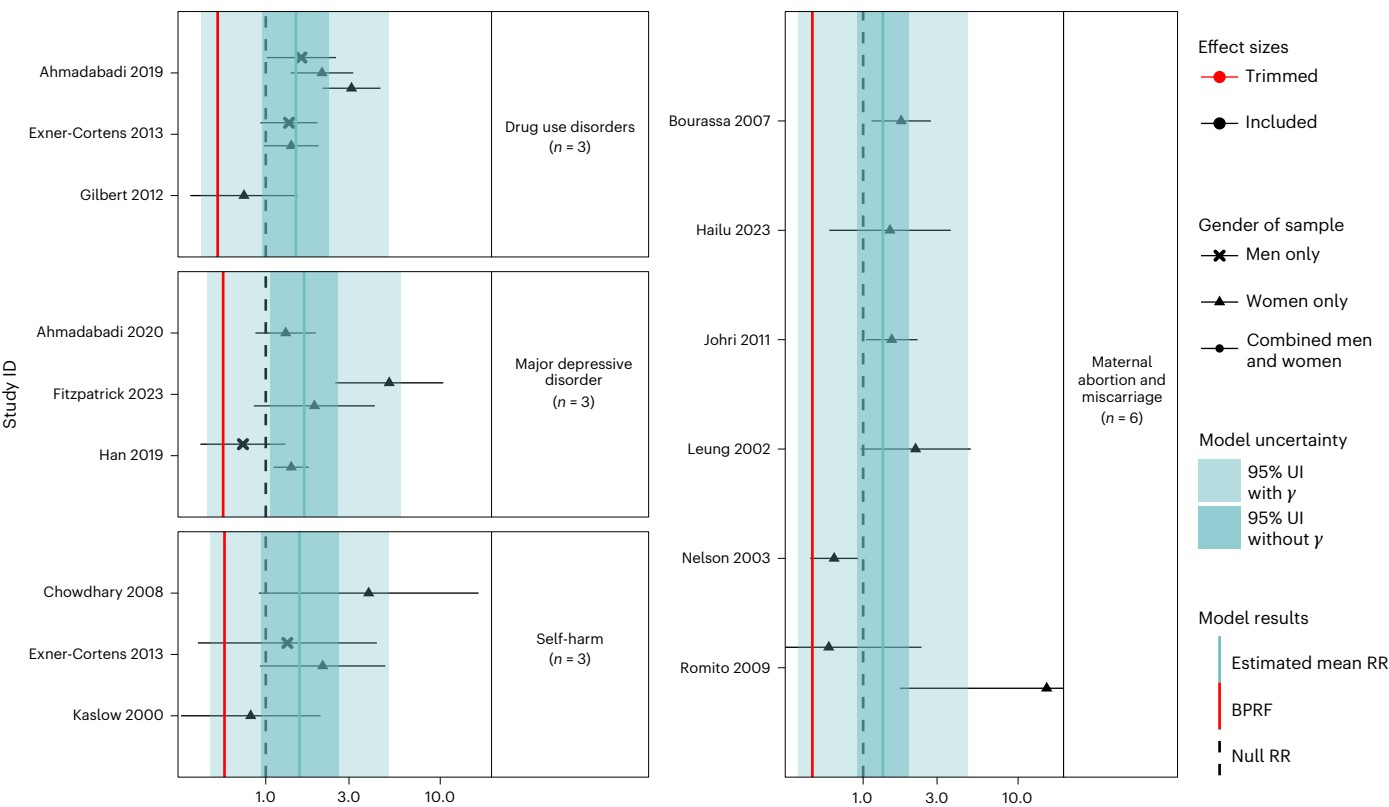

**Fig. 4 | Forest plots for psychological GBV and health outcomes identified through a systematic review of the literature.** The shape of each point indicates the gender of the sample (women only, men only, and combined women and men). Light blue shading corresponds to the 95% UI incorporating between-study heterogeneity; dark blue shading corresponds to the 95% UI without between-study heterogeneity. The black vertical dashed lines reflect the null RR value (one), and the red vertical lines show the BPRF at the fifth quantile for these harmful risk–outcome associations. The black data points and horizontal lines each correspond to an effect size and 95% UI from the included study identified on the y axis. We included multiple observations from a single study when effects were reported by gender, exposure definition and age group. Supplementary Table 1 contains more details on the observations included from each study.

necessary to further our understanding of GBV's full breadth of health consequences. Moreover, data related to various conditions—such as breast cancer, Alzheimer's disease and other dementias, dermatitis, diabetes mellitus, eating disorders, gynaecological diseases, headache disorders, ischemic heart disease, schizophrenia, and multiple sclerosis—and their associations with GBV were available but were reported in fewer than three studies, precluding inclusion in our analysis. Similarly, some evidence and clinical reports also indicate that physical GBV can lead to traumatic brain injuries[69], but this association was not feasible to study given the existing data landscape. The limited availability of high-quality studies on the health challenges of GBV exposure obscures our understanding of GBV's total health burden and limits the development and implementation of nuanced response strategies.

Despite these data gaps, our findings indicate that sexual and physical GBV are consequential risk factors for mental health disorders, substance use disorders, and maternal and reproductive health outcomes, and merit similar degrees of public health attention as is afforded other behavioural risk factors, such as smoking and secondhand smoke[66,67]. Both preventing GBV and mitigating its negative health effects have the potential to markedly reduce the global burden of these widespread conditions. Furthermore, beyond the health implications presented here, exposure to GBV has extensive social and economic repercussions, affecting individuals' well-being and that of families, communities and societies[70]. GBV perpetrated at home, for example, may occur in front of children and has been conceptualized as a form of psychological violence against children (VAC) with its own future health consequences[71]. These vast and generational harms

can be averted through multifaceted, ongoing violence prevention programmes that entail increasing awareness, advancing educational initiatives, and changing social norms that undergird the disturbingly widespread acceptance of GBV[72]. These types of evidence-based, structural approaches have been shown to be successful in decreasing the incidence of violence against both children and women[73].

Simultaneously, it is crucial to bolster health services working with survivors to address the negative consequences post-GBV exposure. Access to quality health care remains a challenge in many regions, and survivors of GBV, in particular, often face additional obstacles in accessing STI and HIV/AIDS care, family planning resources, and appropriate nutrition[57,58,74]. While considerable strides have been made to improve recognition of and care for survivors of GBV, systematic barriers continue to deter health-care providers from identifying and managing GBV cases, including a lack of preparedness and absence of training in responding to GBV and, in particular, IPV[75]. Multi-sectoral interventions, such as improved GBV screening and referral to trauma-informed care, can make an important difference by prioritizing safety, patient autonomy, shared decision-making and empowerment[5,76]. This approach has been effective in reducing negative health outcomes among women who experienced IPV[5,76]. As research continues to advance our understanding of the health consequences of GBV, it is crucial that policymakers, clinicians and funders step up to address the known health needs of existing survivors and strengthen violence prevention efforts to mitigate GBV exposure for future generations.

The present study must be considered within its limitations. First, there was substantial between-study variation in the definitions of GBV

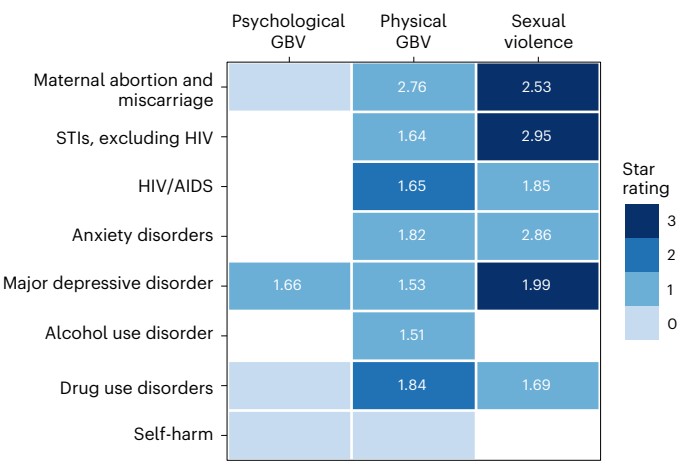

**Fig. 5 | Mean RR and strength of the evidence for the association between multiple forms of GBV and eight health outcomes.** The number in each box corresponds to the mean RR estimated as part of this study for associations for psychological, physical and sexual GBV and specific health outcomes. The coloured boxes represent all associations supported by at least three published studies, allowing for our Burden of Proof assessment, while empty white cells reflect risk–outcome pairs for which we did not have enough data (three or more studies) to examine. The shades of the blue boxes represent the strength of evidence supporting each association based on our conservative interpretation of the data that aligns with the Burden of Proof approach. The strength of the evidence is translated into a star rating from one to five stars on the basis of thresholds outlined in Zheng et al.[14], where one star denotes weak evidence, and each additional star indicates progressively stronger evidence. An absence of stars (zero stars) signifies insufficient evidence of a significant association between the exposure and the outcome.

operationalized, even within those examining the same type of GBV. Variability occurred in relation to recall period (lifetime or past-year exposure), survey mode and perpetrator identity. For example, while our study aimed to examine the health consequences of GBV regardless of the perpetrator, the data for three risk–outcome pairs analysed focused solely on IPV-related forms of GBV, which may affect the generalizability of the findings to GBV perpetrated by non-intimate partners. When possible, we constructed and tested binary bias covariates to control for these differences in exposure definitions, including perpetrator types. However, exposure definitions probably remain an important source of unexplained between-study heterogeneity (captured by γ) due to the large deviations in exposure definitions and the low number of studies overall, which restricted our ability to test bias covariates in some models.

Similarly, we limited our existing analyses to studies that reported distinct effect sizes for physical, sexual and/or psychological GBV to parse out the unique health consequences of each of these types of GBV. However, most studies did not explicitly state whether the exposure definition they used was restricted to individuals who had experienced only the GBV type of interest, or whether it potentially included individuals who had experienced other types of GBV as well. In the latter case, the resulting effect size may overestimate the distinct health effect of the single GBV type, particularly since individuals who experience one form of violence are more likely to experience other forms[77,78]. Studies also very rarely clarified whether the reference group of unexposed individuals excluded or included people who had experienced other forms of GBV, which—conversely—may result in underestimating effect sizes. In the absence of more detailed reporting on the part of included studies, any adjustment or bias covariate to account for other forms of violence would require substantial assumptions on the part of the research team, which may, in turn, introduce additional sources of uncertainty to the model. When input studies provided information

regarding co-occurrence with other forms of violence, we prioritized data points that were limited to groups exposed only to the GBV type of interest and reference groups with no other violence exposure. We also prioritized observations that controlled for any other form of violence to reduce the interference of ill-defined exposure groups in order to focus specifically on the health consequences of physical, sexual or psychological GBV. However, in the interpretation of our findings, it is important to consider that different types of GBV rarely occur independently. Our estimates therefore probably represent the lower bound of risk experienced by an individual exposed to any of these forms of GBV, with co-occurrence leading to compounding health consequences.

A further limitation is the use of dichotomous case definitions for physical, sexual and psychological GBV. This approach does not allow consideration of how differences in frequency or severity may interact with the health consequences of exposure to GBV. However, related to the research gaps discussed above, few of the studies we included reported effect sizes for severity- or frequency-stratified exposed groups relative to an unexposed group. Where this information was available, the categories were varied, preventing meaningful comparison. Furthermore, as exemplified by the paucity of data on male-specific observations in addition to an over-representation of data from high-income countries, the included studies largely reflected similar populations of women in high-income countries and did not provide further layers of granularity to the data (see Supplementary Table 1 for relevant metadata). Given that different sub-populations may experience different forms of violence at different rates or severity levels[77,79,80], future research on the frequency- and severity-related health effects of GBV exposure and among sub-groups, particularly non-binary individuals, is crucial.

Finally, while our analysis excluded studies that exclusively focus on violence that occurred during childhood, we did examine GBV exposure across multiple recall types (lifetime exposure, exposure in the recent past and exposure during pregnancy) and included studies that present results for broad age ranges (including adolescent, young adult and adult participants). As a result, we included several studies that assessed exposure to violence across multiple recall periods and/ or did not explicitly state that the violence resulted from unequal gender-power relations (or the motivation of violence). Consequently, we were not always able to clearly distinguish GBV from other forms of violence, including VAC. While these studies were included if they contained data related to our exposure definitions of GBV, it is difficult to clearly disentangle the effects of childhood violence or other forms of violence from those of GBV. This challenge may muddle the GBV interpretation of our results, despite excluding studies that solely focus on children.

A concurrent analysis based on the same systematic review is examining the health effects of physical abuse, psychological abuse and neglect against children below the age of 18, and a previously published Burden of Proof analysis using prior iterations of the systematic review focused on childhood sexual abuse and on physical and/or sexual IPV above the age of 15 (ref. 13). Together, these works present a broader view of the health consequences of GBV and VAC across the lifespan. However, it is important to highlight the role that GBV experienced during adolescence, particularly non-partner-perpetrated forms of GBV, may play in the overall health toll of GBV. Adolescence is a critical life stage for physical and social development and is an important period for laying the foundations of good health. While the current studies include adolescents in their results, exposure to GBV during this pivotal developmental stage may result in unique health outcomes, many of which could be masked in the broader analysis. GBV during adolescence therefore merits further focused investigation as an impactful risk factor for future health.

In this systematic review and meta-analysis, we examined the detrimental health effects of physical, sexual and psychological GBV.

Our findings highlight the fact that GBV is a pressing risk factor of global health significance. Our analyses identified moderate and weak evidence of associations between physical, sexual or psychological GBV and eight health outcomes, highlighting the need to advance the evidence base on the health consequences of GBV. Furthermore, we identified crucial data gaps in the existing literature on GBV among men and on psychological GBV as a distinct health risk factor. Beyond the necessity of recognizing GBV as a global health research priority, there is a further need to prevent and mitigate the health harms of GBV in current and future generations. An integrated approach—at community, regional, national and global levels, including actors from society, health services, educational and judicial systems, and law enforcement, along with policymakers and scientists—is needed to eradicate GBV and provide the support urgently needed by survivors.

## Methods

### Overview
In this study, the Burden of Proof methodology was applied to estimate the association between exposure to physical, sexual and psychological GBV (modelled as dichotomous risk factors) and selected health outcomes and to evaluate the strength of evidence underlying the estimates of association. We applied the Burden of Proof methodology if a risk–outcome pair had at least three studies identified in the scientific literature. We generated estimates of RR, BPRF and ROS for all risk–outcome pairs in a single model with no location- or age-specific results. The estimates reflect input data predominately on women, but also include input data for men and combined for men and women, drawing upon all available data regardless of how or whether the input study collected and reported data by sex or gender. Given that both sex and gender data were used in the underlying studies, we opted to use more inclusive gender-specific terminology (men/women) in our results and interpretation.

Zheng and colleagues established the methods that generate the BPRF[14], which use the meta-analytic tool MR-BRT (meta-regression— Bayesian, regularized, trimmed). Published studies on other risk factors—red meat consumption[81], smoking[67], vegetable consumption[82], high systolic blood pressure[83], and IPV against women and child sexual abuse[13]—have used the same approach, following six main analytical steps: (1) conducting a systematic review of the literature and data extraction of all included studies; (2) conducting a meta-analysis to estimate the RR of the health outcome occurring as a function of exposure to the risk factor (compared with no exposure to the risk factor) for each risk–outcome pair; (3) examining and adjusting for systematic sources of bias related to known heterogeneity in input study-design characteristics; (4) quantifying remaining between-study heterogeneity ($\gamma$) using random-effects modelling and incorporating this value into UIs around the mean RR; (5) assessing publication bias using Egger's regression test and visual inspection of funnel plots; and (6) estimating the BPRF to compute the risk increase due to exposure according to available data and to produce the ROS, which is further mapped onto five star-rating categories of risk.

This study complies with the Guidelines on Accurate and Transparent Health Estimate Reporting (GATHER)[84] recommendations (Supplementary Table 9). We followed the PRISMA (Preferred Reporting Items for Systematic Reviews and Meta-analyses)[85] guidelines through all stages of this study (Supplementary Tables 10 and 11). The Burden of Proof methodology, as a component of the Global Burden of Disease (GBD) study, was approved by the University of Washington Institutional Review Board (study no. 9060), and the systematic review approach was registered in PROSPERO (CRD42022299831).

### Systematic review and meta-analysis
In partnership with the Lancet Commission on Gender-Based Violence and Maltreatment of Young People, we carried out a systematic review to identify studies reporting on exposure to GBV and VAC and their relationship with any health outcomes. The review was carried out in accordance with a prospectively published review protocol[12]. In brief, we searched for studies published from 1 January 1970 to 31 January 2024, without language restriction, in seven databases: PubMed, Embase, CINAHL, PsycINFO, Global Index Medicus, Cochrane and Web of Science Core Collection. The search string included keywords aimed at restricting studies on the basis of (1) violence exposure, (2) study design and type, (3) measures of association and/or risk, and (4) publication year (Supplementary Information section 4). There were no limitations placed on the types of outcomes to capture all potential health outcomes that have been studied in relation to GBV or VAC exposure. During screening and analysis, we referred to a reference list of health outcomes spanning all those included in the GBD[13,86].

Using the systematic review software Covidence, the first phase of study selection after deduplication of records was identifying through title/abstract review whether a study may report on the relationship between exposure to GBV or VAC and health outcomes. The titles and abstracts of 75,331 identified articles were screened by a group of trained reviewers. The first two thirds of the titles and abstracts were reviewed by two independent reviewers, and any disagreements were resolved by a third reviewer. At this point, less than 5% of screened studies had conflicting decisions, so the remainder of the titles and abstracts were single screened. The full text of 4,446 studies, and an additional 312 studies identified through systematic review citation searching, were then screened to confirm their inclusion/exclusion in our final dataset. To merit inclusion, studies needed to (1) use an eligible study design (cohort, case–control or case–crossover) that allowed the research team to determine temporality between the violence exposure and development of the health outcome, (2) report a measure of association or enough detail to derive a measure of association between GBV or VAC exposure and a health outcome, and (3) appropriately define the exposure and outcomes. For example, composite measures of violence including explicitly non-gender-based violence cases were not eligible for inclusion. Studies were excluded if they (1) used cross-sectional, ecological, case series or case study designs; (2) failed to establish temporality; or (3) reported incomplete data. More details on the inclusion/exclusion criteria can be found elsewhere[12,13] and in Supplementary Tables 12 and 13. Ultimately, 578 studies were identified reporting the health impacts of any form of GBV and/or VAC. An additional 57 studies were identified through citation searching systematic reviews/meta-analyses for a total of 635 studies.

Study characteristics were extracted using a modified Covidence v.2.0 extraction template (Supplementary Table 14) and included author, year, study design, age, gender, sample size, number of cases exposed, number of cases unexposed, violence type included in the exposed definition, outcome definition, perpetrator type, effect sizes and confidence intervals, and sources of potential bias. Details on all the information included in the extraction template can be found in Supplementary Table 1.

### Defining exposures of interest
While the systematic review captured all forms of GBV or VAC, the current analysis focused on distinct forms of GBV: physical GBV, psychological GBV and sexual violence. For the purposes of the present analysis, we defined physical GBV as a deliberate, unwanted and non-essential act of physical force against an individual's body due to aspects of their identity related to gender. Similarly, psychological GBV was defined as deliberate, unwanted and non-essential verbal or non-verbal acts driven by gender-related components of the victim's identity that result in long-term psychological harm. These acts can include terrorizing, harassing, spurning, humiliating and controlling. Sexual violence, considered inherently a form of GBV, is any deliberate, unwanted and non-essential sexual act, including both completed and attempted rape, sexual assault, and non-contact sexual acts. The definitions operationalized here are based on and adapted from similar

categories proposed in the International Classification of Violence Against Children[71] but have a narrow focus on specific forms of GBV.

For this study, we analysed violence exposure across multiple recall types, including studies that captured lifetime exposure to GBV, exposure to GBV during adulthood and exposure to GBV within specific time periods, including in the past three years, in the past year, in the past six months, in the past three months and during pregnancy. For studies that spanned childhood through adulthood, the age of participants and exposure timing were taken into account to evaluate whether the study aligned with our exposure definitions of GBV. We did not include studies that exclusively examined the health effects associated with physical abuse, psychological abuse and neglect that occurred during childhood in this analysis, as these forms of violence are considered separate from GBV and are reported in a concurrent study using the aforementioned systematic review[87].

A previously published analysis examined the health consequences of childhood sexual abuse, as well as exposure to physical and/or sexual IPV against women[13] (Supplementary Table 15 and Supplementary Fig. 22). While our analyses do not focus on violence perpetrated by partners, physical IPV and sexual IPV are considered sub-groups of physical GBV and sexual violence, respectively. However, our study complements this prior analysis by (1) including more recent literature, (2) examining psychological GBV, (3) parsing out the distinct effects of physical GBV and sexual violence rather than their combined exposure, and (4) incorporating both partner-perpetrated GBV and violent acts perpetrated by non-partners because of an individual's sex, gender identity, gender expression, or expression of masculinity or femininity. Furthermore, our analysis is not limited by the gender of the survivor of violence and includes GBV perpetrated against men. However, due to a paucity of data distinguishing the motivations of physical or psychological violence perpetrated against men due to their sex or gender, rather than perpetrated against men due to other factors, the scope of the studies on physical or psychological GBV against men, specifically, is largely limited to physical or psychological IPV. In comparison, the studies capturing physical or psychological GBV against women that were included in the present analysis have a broader range of perpetrators.

## Selection of data on sexual, physical and psychological GBV and its health outcomes

The present analysis leverages the studies included in the aforementioned systematic review to identify the subset with data on the health effects of sexual, physical and psychological GBV. Studies were eligible for the current analysis if they reported on the association between a health outcome and one of the three exposure types of interest. Studies that reported a measure of association for a health outcome and a composite form of GBV (for example, studies in which exposure was defined as sexual and/or physical GBV without reporting different effect sizes or sample sizes for each) were not eligible for the current violence-type-specific analyses even if they were included in the systematic review. If a study provided multiple effect sizes for the same type of GBV exposure for the same health outcome, we prioritized observations derived from analytical samples limited to individuals who had experienced only the violence type of interest to reduce the impact of potential co-occurrence of other forms of GBV. In lieu of such explicit restrictions to the exposed group, we prioritized effect sizes that were adjusted for exposure to at least one other form of violence. Our reference exposure definitions (Supplementary Table 16) also refer to exposure to violence types of interest at any point, although we also accepted observations for exposure to violence types of interest during other time periods (for example, in the past year) or variations of the reference exposures (Supplementary Table 17). For studies that reported effect sizes for the same health outcome and the same exposure type but during multiple recall periods, we prioritized the data points with the longest recall, with a particular emphasis on lifetime exposure.

Among the studies that reported on an eligible form of GBV, we identified the health outcomes of interest within each violence type as the GBD causes of health burden that had been studied in three or more published reports. In other words, we identified the health outcomes for each type of GBV with the minimum number of data points necessary to appropriately analyse. Studies that reported effect sizes only for combinations of GBD causes were not eligible for the present analyses. Studies that reported effect sizes of more granular health outcomes than those used in our current analyses are included in their corresponding health outcome of interest (for example, cannabis use disorder is categorized in drug use disorders, and syphilis is categorized in STIs). If a study reported multiple effect sizes for the same violence types and outcomes at different levels of granularity, we selected the effect sizes with the outcome definition that most closely matched the GBD definitions. We conducted sensitivity analyses omitting various alternative outcome definitions when relevant and feasible, which are described in more detail in Supplementary Information sections 6.1.1 and 6.1.2. More details on identified outcomes of interest and the corresponding accepted outcome definitions can also be found in Supplementary Tables 18 and 19.

Using these criteria, we identified 40 unique eligible studies that covered eight health outcomes with sufficient data to examine their association with exposure to physical GBV, six health outcomes for sexual violence and four for psychological GBV. Studies with more than one eligible observation related to a given GBV type and health outcome were further selected on the basis of perpetrators and analytical sample. Namely, our primary analysis aimed to examine the health effects of GBV, regardless of the perpetrator. We therefore selected the broadest perpetrator type for the studies that reported both effect sizes stratified by perpetrator and effect sizes for GBV regardless of the perpetrator. When possible, we also conducted separate additional analyses focused solely on IPV and including only observations with GBV from unrestricted or unspecified perpetrators. There were no health outcomes or violence types with sufficient data on explicitly non-partner-perpetrated GBV to conduct separate non-partner analyses. After we made these selections, remaining multiple observations from the same study were further selected to prioritize those that were not sub-group analyses and that were derived from analytical samples of combined men and women participants, as opposed to gender-stratified effect sizes. We conducted further gender-specific sensitivity analyses in which the gender-stratified effect sizes were selected for inclusion instead to identify differences in risk by gender. Any further studies with multiple observations underwent a manual vetting process to account for more granular differences. To account for studies in which multiple observations from the same analytical sample met our selection criteria of closest exposure and outcome definitions to our reference definitions, least restrictive perpetrator type, and broadest sample grouping, we applied a standard error inflation factor of the square root of the number of observations for the same risk–outcome pair derived from non-mutually-exclusive age, gender and location study samples. This inflation factor is an approach to reduce the impact of a single study in our models in the absence of further data on the degree of overlap in the participants informing different effect sizes.

## Testing and adjusting for biases across study designs and characteristics

We followed the Grading of Recommendations, Assessment, Development and Evaluations (GRADE)[88] to create binary bias covariates based on the risk-of-bias assessment criteria: (1) method of exposure measurement (instrument or survey used) and data source (self-reported versus ascertained from administrative sources of information such as legal or health-care databases), (2) method of measuring the outcome (instrument, survey or diagnostic criteria used) and data source (self-reported versus ascertained from administrative sources), (3) representativeness of the study population, (4) control for confounding, (5) risk of

selection bias (considering the percentage of follow-up for longitudinal studies and percentages of cases and controls for exposed groups that could be ascertained for case–control designs), and (6) reverse causation (assessed through study design and recall bias (that is, case–control studies)). Covariates included whether the sample was representative of the underlying location, whether the study was at risk for selection bias with loss to follow-up (cohorts) or percentage without ascertained data (case–controls) >20%, confounding uncontrolled if the study reported only a completely unadjusted effect size, and non-lifetime recall if the exposed definition was past-year exposure to the violence type or a shorter time period. The complete list of covariates created can be accessed in detail in Supplementary Tables 20–23, and the bias covariates marked and tested for each risk–outcome pair are provided in Supplementary Tables 24–26.

Additionally, some covariates were specifically created for exposure to GBV, as follows: whether the perpetrator was limited specifically to partners or any perpetrator, whether only one gender was included in the study (that is, women only or men only), and pregnancy recall if the exposure to violence occurred during pregnancy (Supplementary Table 22). Bias covariates were also created regarding the health outcomes; we consulted cause-specific research teams at the Institute for Health Metrics and Evaluation to verify and evaluate cause definitions and best practices for measuring the relevant health outcome and whether or not outcome measurement methods were acceptable (for example, the use of diagnostic interview versus symptom scale for measuring major depressive disorder) (Supplementary Tables 19 and 23).

We used MR-BRT's automated covariate selection process to identify and adjust for statistically significant bias covariates[14]. The technique uses a Lasso strategy to evaluate and sequentially rank potential bias covariates created to reflect known variation across input study characteristics. Bias covariates were added individually, according to rank, to a linear meta-regression model and assessed on the basis of significant association with effect size, with the process terminated as soon as a covariate was added that did not have a significant effect. This technique requires at least two rows of data for each value of the covariate (0 and 1). In some cases, due to the smaller number of studies included in a meta-analysis of risk–outcome pairs, not all covariates created were feasible to test, as this criterion was not met (Supplementary Information section 7.2). The significant bias covariates were included in the final mixed-effect regression model.

### Quantifying between-study heterogeneity

To evaluate the consistency of findings across the available literature, we further used random-effects modelling (for example, $\gamma$ and its uncertainty) instantiated in the MR-BRT tool to quantify the heterogeneity between studies that remained after accounting for known heterogeneity using covariate selection and adjustment. A study-level random slope was added to the final linear fixed-effects model to estimate $\gamma$, and the uncertainty of $\gamma$ was assessed using the inverse Fisher information matrix to account for small study effects. We incorporated the 95th lower and upper bounds of $\gamma$ into conventionally derived posterior fixed-effect UIs around mean RR estimates to generate UIs with between-study heterogeneity($\gamma$)[14]. In this study, we report pooled RR estimates alongside both conventional 95% UIs without $\gamma$ and updated 95% UIs with $\gamma$. The former are presented to facilitate comparison with other meta-analyses, while the latter reflect RR in addition to the degree of consistency within the existing literature and the quantity of existing evidence to better contextualize pooled RR estimates and to provide the basis for the BPRF measure.

### Evaluating publication bias

Using the Egger regression test[89], we assessed publication and reporting bias in the input data. This regression evaluates whether there is a significant correlation between the residuals and their standard error and complements a visual inspection of the funnel plot asymmetry

(Supplementary Figs. 23–40). In the present study, no publication bias was found in the risk–outcome pairs evaluated.

### Estimating the BPRF

All the risk–outcome pairs presented in this study are dichotomous pairs in which the risk of the outcome is compared between individuals who are exposed to the risk factor versus a reference group that is unexposed, regardless of the frequency or severity of exposure (Supplementary Table 27). In this context, the BPRF reflects the most conservative estimate of the harmful association between exposure to GBV and the selected health outcome, consistent with available evidence. It is estimated as the fifth quantile of draws closest to the null from the distribution defined by the RR UIs inclusive of between-study heterogeneity[14]. In this manner, the BPRF incorporates both the degree of certainty in the point estimate and the underlying variation in the data. It can be conceptualized as an increase in the risk of the disease outcome by at least the value of (BPRF − 1) × 100 among exposed individuals and can be transformed into a ROS of log(BPRF)/2. A higher positive ROS means that a relationship between exposure and outcome is characterized by a larger effect size and/or strong underlying evidence, while a lower and negative ROS indicates weak evidence of a relationship.

To make ROSs easier to interpret by policymakers and research funders and easier to compare between dichotomous and continuous risk–outcome pairs, ROS values are mapped onto a star-rating system from one to five stars on the basis of pre-established thresholds. The absence of a star rating (zero stars) indicates that the lower bound of the RR 95% UI without between-study heterogeneity, reflecting conventional measures of association, crosses the null RR value of one. In other words, the absence of a star rating indicates no credible evidence of an association between the exposure and the outcome. For these pairs, the ROS is not calculated as we cannot evaluate the strength of evidence underlying no association, and the risk–outcome pair is not eligible for inclusion in the GBD. From there, a one-star rating denotes weak evidence of association, and a five-star rating indicates strong evidence of an association between exposure to a given risk factor and a specific health outcome.

### Model validation

The methods used to generate the BPRF and ROS were developed and validated by Zheng and colleagues[14]. In addition to the main models, we ran sensitivity analyses considering the type of perpetrator (studies that did not specify a perpetrator or only examined IPV) and the gender of participants (only women or only men), and excluding pregnancy recall when we had enough studies to do so. Across all risk–outcome pairs for which our input modelling dataset was ≥10 observations, we undertook a sensitivity analysis in which we did not apply 10% trimming (trimming removes points that are least coherent with most of the data). We additionally conducted several outcome-specific analyses in which we evaluated the impact of excluding studies with certain outcome characteristics identified a priori or via bias covariate selection (Supplementary Information section 2).

### Reporting summary

Further information on research design is available in the Nature Portfolio Reporting Summary linked to this article.

## Data availability

The findings from this study are supported by data extracted from published literature databases (PubMed, Embase, CINAHL, PsycINFO, Global Index Medicus, Cochrane and Web of Science Core Collection). Citations for all input data are provided as part of this manuscript. Study characteristics and all included data points are provided in the Supplementary Information (Supplementary Tables 1 and 28). Details on data sources can also be found on the GHDx website (https://ghdx.healthdata.org/record/ihme-data/gbv-health-effects-bop-risk-outcome-scores).

## Code availability

All code used for these analyses is publicly available online (https://github.com/ihmeuw-msca/burden-of-proof/). The analyses were carried out using R version 4.0.5 and Python version 3.10.9. The MR-BRT tool was used to generate the BPRF.

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

## Acknowledgements

Research reported in this publication was supported by the Bill & Melinda Gates Foundation (award no. 66-7813, E.G.). The funders had no role in study design, data collection and analysis, decision to publish or preparation of the manuscript.

## Author contributions

A.A., M.J.B.d.R., J.C., R.J.C.C.-A., S.C., J.K.C., C.V.N.C., G.N.d.A., F.M.D.d.A., B.H., M.H., M.K., R.Q.H.L., S.M., J.K.M., V.P., S.R., D.S., C.N.S., A.T., A.V., G.F.G., C.S. and N.V. were primarily responsible for seeking, cataloguing, extracting or cleaning data; and designing or coding figures and tables. A.A., J.C., R.J.C.C.-A., J.S.C., F.M.D.d.A., A.N.D., L.S.F., E.G., B.H., F.M.K., R.Q.H.L., N.M., V.P., D.S., R.J.D.S., C.S., C.N.S., G.F.G., H.S., A.V. and N.V. provided data or critical feedback on data sources. A.Y.A., F.B., J.S.C., R.J.C.C.-A., L.S.F., E.G., S.I.H., F.M.K., N.M., C.J.L.M., L.S.F., C.N.S., C.S., G.F.G., H.S. and P.Z. provided critical feedback on methods or results. F.B., J.S.C., E.G., S.I.H., F.M.K., C.J.L.M., S.A.M., N.M., C.N.S., C.S., L.S.F., G.F.G., E.D. and H.S. drafted the work or critically revised it for important intellectual content. E.G., E.C.M., E.M.O., C.S., E.D. and C.N.S. managed the overall research enterprise. A.Y.A., S.C., C.J.L.M., R.J.D.S. and P.Z. developed the methods or computational machinery. C.S. and G.F.G. were primarily responsible for applying analytical methods to produce estimates. E.G., J.S.C., N.M., C.N.S., L.S.F., G.F.G. and C.S. wrote the first draft of the manuscript. E.G., S.I.H., E.C.M., E.M.O., L.S.F. and C.S. managed the estimation or publication process.

## Competing interests

G.F.G. reports grants or contracts from Bloomberg Philanthropies with payment through salary at IHME. F.B. reports support for attending meetings and/or travel from Partnership for Maternal, Newborn and Child Health, Geneva, Switzerland, and Fondation Botnar, Basel, Switzerland; leadership or fiduciary role in other board, society, committee or advocacy group, paid or unpaid as Chair of Governance and Ethics Committee for the Partnership of Maternal, Newborn and Child Health; International Advisory Board Chair of United Nations University International Institute for Global Health, Kuala Lumpur, Malaysia; Co-Chair of the Lancet Commission on Gender Based Violence and the Maltreatment of Young People; Vice Chair Fondation Botnar, Basel, Switzerland; Member of the Lancet Future of Neonatology Commission; and Member of the Lancet and Chatham House Commission on Universal Health Member of Lancet Commission on Investing in Health 3. J.K.C. reports grants or contracts from the University of Warwick and support for attending meetings and/or travel from the University of Miami and the University of Washington. J.S.C. reports grants or contracts from the National Institute for Health and Care Research, Youth Endowment Fund, College of Policing, University of Birmingham and Birmingham City Council; and support for attending meetings and/or travel from the University of Miami and the University of Washington. B.H. reports grants or contracts from West Midlands Secure Data Environment (£50,000 Pump Priming Fund for project titled 'Developing an automated evaluation pipeline to identify the effectiveness of digital interventions in acute care: a pilot study assessing inequalities in the effectiveness of DERM'). F.M.K. reports grants or contracts from Merck KGaA/EMD Serono (research grant to the University of Miami); Tides Foundation via the Oak Foundation (two research grants to the University of Miami); Fondation Botnar (research grant to the University of Miami); Finker-Frenkel Family Foundation (gift to the University of Miami to support the Lancet Commission on Gender-Based Violence and Maltreatment of Young People); Wellcome Trust (gift to the University of Miami to support the Lancet Commission on Gender-Based Violence and Maltreatment of Young People); Mena Catering (gift to the University of Miami to support the Lancet Commission on Gender-Based Violence and Maltreatment of Young People); Gloria Estefan Foundation (gift to the University of Miami to support the Lancet Commission on Gender-Based Violence and Maltreatment of Young People); and Jose Milton Foundation (gift to the University of Miami to support the Lancet Commission on Gender-Based Violence and Maltreatment of Young People). F.M.K. also reports consulting fees from Merck KGaA/EMD Serono (personal consulting agreement to advise the company's research/dissemination strategy for 'Healthy Women, Healthy Economies' and 'Embracing Carers' initiative focused on caregiving and women in leadership; totally unrelated to the subject of this paper/no work related to child sexual abuse) and Tecnológico de Monterrey (provide strategic guidance on research priorities and lectures for the Institute for Obesity Research at the Tecnólogico de Monterrey (university); totally unrelated to the subject of this paper/no work related to child sexual abuse); and leadership or fiduciary role in other board, society, committee or advocacy group, paid or unpaid from Founding President, Tómatelo a Pecho, A.C. (a Mexican non-profit organization that has promoted research, advocacy, awareness and early detection of breast cancer since its inception, and has since expanded to promote women's and girls' health broadly and health systems); Esperanza United (Member, Board of Directors—unpaid); and Senior Economist, Mexican Health Foundation (unpaid). The other authors declare no competing interests.

## Additional information

**Extended data** is available for this paper at https://doi.org/10.1038/s41562-025-02144-2.

**Correspondence and requests for materials** should be addressed to Emmanuela Gakidou.

Caroline Stein [1,2], Luisa S. Flor[1,2], Gabriela F. Gil[1], Mariam Khalil[1], Molly Herbert[1], Aleksandr Y. Aravkin[1,3], Alejandra Arrieta [1,2], María Jose Baeza de Robba[4,5], Flavia Bustreo[6,7], Jack Cagney [1], Renzo J. C. Calderon-Anyosa[8], Sinclair Carr [1], Jaidev Kaur Chandan[9,10], Joht Singh Chandan [10], Carolina V. N. Coll[11,12], Fabiana Martins Dias de Andrade[13], Gisele N. de Andrade [13], Alexandra N. Debure [14], Erin DeGraw[1], Ben Hammond [10], Simon I. Hay [1,2], Felicia M. Knaul[15,16,17,18], Rachel Q. H. Lim[10], Susan A. McLaughlin[1], Nicholas Metheny[19], Sonica Minhas [10], Jasleen K. Mohr[10], Erin C. Mullany[1], Christopher J. L. Murray [1,2], Erin M. O'Connell[1], Vedavati Patwardhan[20], Sofia Reinach [21], Dalton Scott [14], Cory N. Spencer[1], Reed J. D. Sorensen[1], Heidi Stöckl [22], Aisha Twalibu[1,2], Aiganym Valikhanova[1], Nádia Vasconcelos[1,13], Peng Zheng[1,2] & Emmanuela Gakidou [1,2] ✉

[1]Institute for Health Metrics and Evaluation, University of Washington, Seattle, WA, USA. [2]Department of Health Metrics Sciences, School of Medicine, University of Washington, Seattle, WA, USA. [3]Department of Applied Mathematics, University of Washington, Seattle, WA, USA. [4]School of Nursing, Pontifical Catholic University of Chile, Santiago, Chile. [5]Center for Global Health Equity, University of Michigan, Ann Arbor, MI, USA. [6]Fondation Botnar, Basel, Switzerland. [7]Partnership for Maternal, Newborn and Child Health, Geneva, Switzerland. [8]McGill University, Montreal, Quebec, Canada. [9]Warwick Medical School, University of Warwick, Coventry, UK. [10]Institute of Applied Health Research, University of Birmingham, Birmingham, UK. [11]Department of Epidemiology, Federal University of Pelotas, Pelotas, Brazil. [12]Human Development and Violence Research Center, Federal University of Pelotas, Pelotas, Brazil. [13]Federal University of Minas Gerais, Belo Horizonte, Brazil. [14]School of Nursing and Health Studies, University of Miami, Coral Gables, FL, USA. [15]Institute for the Advanced Study of the Americas, University of Miami, Coral Gables, FL, USA. [16]Department of Medicine, David Geffen School of Medicine, UCLA, Los Angeles, CA, USA. [17]Escuela de Medicina y Ciencias de la Salud, Tecnológico de Monterrey Faculty of Excellence, Mexico City, Mexico. [18]Tómatelo a Pecho, A.C., Mexico City, Mexico. [19]Nell Hodgson Woodruff School of Nursing, Emory University, Atlanta, GA, USA. [20]Center on Gender Equity and Health, UC San Diego School of Medicine, San Diego, CA, USA. [21]Vital Strategies, New York, NY, USA. [22]Institute of Medical Information Processing, Biometry and Epidemiology, Ludwig-Maximilians-University Munich, Munich, Germany. ✉e-mail: gakidou@uw.edu

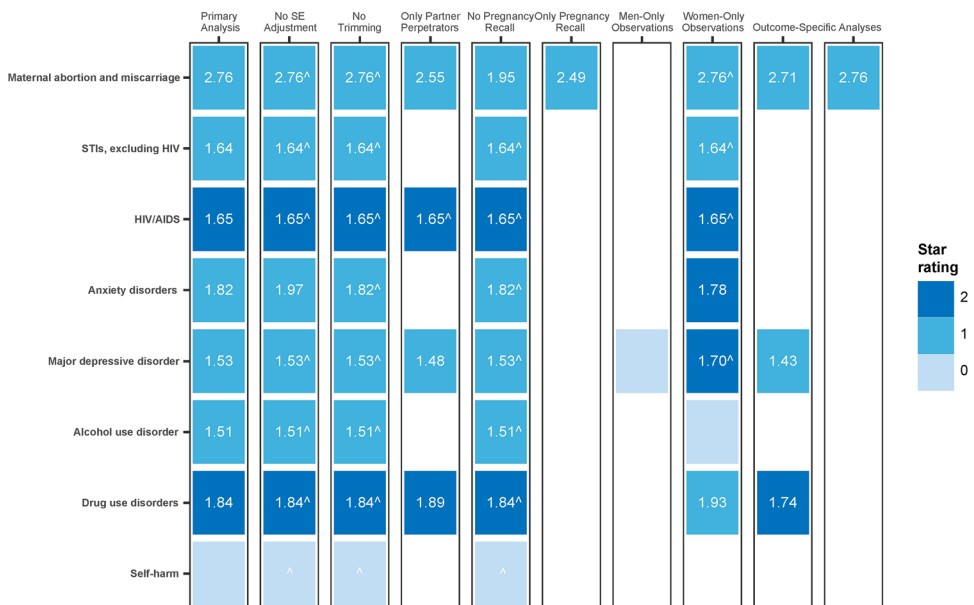

**Extended Data Fig. 1 | Mean relative risk and strength of evidence found in sensitivity analyses on the associations between physical GBV and various health outcomes.** This heatmap compares the estimated mean relative risk and the star rating for the association between physical GBV and each of the seven health outcomes evaluated across different sensitivity analyses. The left most column, titled "Primary analysis," reflects the primary results of the present manuscript. Each consecutive column reflects a different sensitivity analysis (described in more detail in the Methods) specific model restrictions or restrictions to the input data. The cells marked with a ^ reflect a restriction does not change the data or modeling parameters from the primary analysis, resulting in matching models. Each row corresponds to a different outcome of interest. The number overlayed on each cell corresponds to the mean relative risk estimate derived from each analysis for the relationship between the health outcome listed on the y-axis and exposure to physical violence. Colored cells with no overlayed text reflect zero-star risk-outcome pairs in which the analysis in question did not find sufficient evidence of an association between the outcome and physical violence. The cells are colored in accordance with the star rating based on a conservative interpretation of the data in the sensitivity analysis of interest on a 1 (weak evidence) to 5 (strong evidence) star scale. Empty white cells reflect sensitivity analyses for which we did not have enough relevant data (three or more studies) to examine.

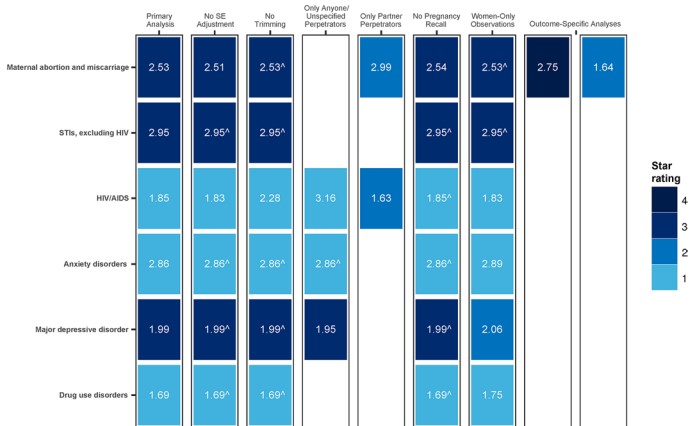

**Extended Data Fig. 2 | Mean relative risk and strength of evidence found in sensitivity analyses on the associations between sexual violence and various health outcomes.** This heatmap compares the estimated mean relative risk and the star rating for the association between sexual violence and each of the six health outcomes evaluated across different sensitivity analyses. The left most column, titled "Primary analysis," reflects the primary results of the present manuscript. Each consecutive column reflects a different sensitivity analysis (described in more detail in the Methods) specific model restrictions or restrictions to the input data. Each row corresponds to a different outcome of interest. The cells marked with a ^ reflect a restriction does not change the data or modeling parameters from the primary analysis, resulting in matching models. The number overlayed on each cell corresponds to the mean relative risk estimate derived from each analysis for the relationship between the health outcome listed on the y-axis and exposure to sexual violence. The cells are colored in accordance with the star rating based on a conservative interpretation of the data in the sensitivity analysis of interest on a 1 (weak evidence) to 5 (strong evidence) star scale. Empty white cells reflect sensitivity analyses for which we did not have enough relevant data (three or more studies) to examine.

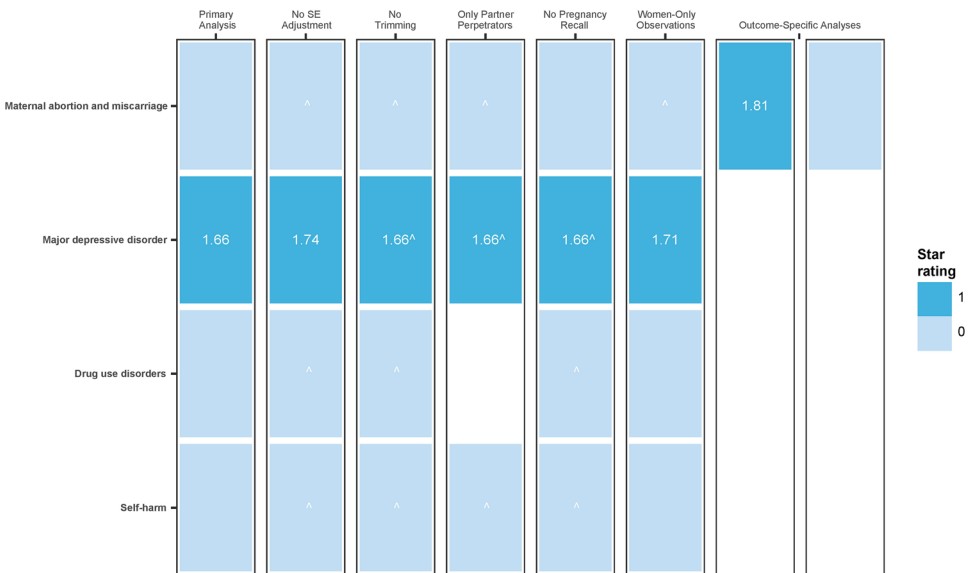

**Extended Data Fig. 3 | Mean relative risk and strength of evidence found in sensitivity analyses on the associations between psychological GBV and various health outcomes.** This heatmap compares the estimated mean relative risk and the star rating for the association between psychological GBV and each of the two health outcomes evaluated across different sensitivity analyses. The left most column, titled "Primary analysis," reflects the primary results of the present manuscript. Each consecutive column reflects a different sensitivity analysis (described in more detail in the Methods) specific model restrictions or restrictions to the input data. Each row corresponds to a different outcome of interest. The cells marked with a ^ reflect a restriction does not change the data or modeling parameters from the primary analysis, resulting in matching models. The number overlayed on each cell corresponds to the mean relative risk estimate derived from each analysis for the relationship between the health outcome listed on the y-axis and exposure to psychological violence. Colored cells with no overlayed text reflect zero-star risk-outcome pairs in which the analysis in question did not find sufficient evidence of an association between the outcome and psychological violence. The cells are colored in accordance with the star rating based on a conservative interpretation of the data in the sensitivity analysis of interest on a 1 (weak evidence) to 5 (strong evidence) star scale. Empty white cells reflect sensitivity analyses for which we did not have enough relevant data (three or more studies) to examine.

# Reporting Summary

## Statistics

For all statistical analyses, confirm that the following items are present in the figure legend, table legend, main text, or Methods section.

| n/a | Confirmed | |
|---|---|---|
| ☐ | ☒ | The exact sample size (*n*) for each experimental group/condition, given as a discrete number and unit of measurement |
| ☐ | ☒ | A statement on whether measurements were taken from distinct samples or whether the same sample was measured repeatedly |
| ☐ | ☒ | The statistical test(s) used AND whether they are one- or two-sided<br>*Only common tests should be described solely by name; describe more complex techniques in the Methods section.* |
| ☐ | ☒ | A description of all covariates tested |
| ☐ | ☒ | A description of any assumptions or corrections, such as tests of normality and adjustment for multiple comparisons |
| ☐ | ☒ | A full description of the statistical parameters including central tendency (e.g. means) or other basic estimates (e.g. regression coefficient) AND variation (e.g. standard deviation) or associated estimates of uncertainty (e.g. confidence intervals) |
| ☐ | ☒ | For null hypothesis testing, the test statistic (e.g. *F*, *t*, *r*) with confidence intervals, effect sizes, degrees of freedom and *P* value noted<br>*Give P values as exact values whenever suitable.* |
| ☐ | ☒ | For Bayesian analysis, information on the choice of priors and Markov chain Monte Carlo settings |
| ☒ | ☐ | For hierarchical and complex designs, identification of the appropriate level for tests and full reporting of outcomes |
| ☐ | ☒ | Estimates of effect sizes (e.g. Cohen's *d*, Pearson's *r*), indicating how they were calculated |

*Our web collection on statistics for biologists contains articles on many of the points above.*

## Software and code

Policy information about availability of computer code

| Data collection | No primary data collection was carried out for this analysis. |
|---|---|
| Data analysis | All code used for these analyses is publicly available online (https://github.com/ihmeuw-msca/burden-of-proof). Analyses were carried out using R version 4.0.5 and Python version 3.10.9. The MR-BRT (meta-regression—Bayesian, regularized, trimmed) tool was used to generate the Burden of Proof Risk Function. |

For manuscripts utilizing custom algorithms or software that are central to the research but not yet described in published literature, software must be made available to editors and reviewers. We strongly encourage code deposition in a community repository (e.g. GitHub). See the Nature Portfolio guidelines for submitting code & software for further information.

## Data

Policy information about availability of data

All manuscripts must include a data availability statement. This statement should provide the following information, where applicable:
- Accession codes, unique identifiers, or web links for publicly available datasets
- A description of any restrictions on data availability
- For clinical datasets or third party data, please ensure that the statement adheres to our policy

The findings from this study are supported by data extracted from published literature databases (PubMed, Embase, CINAHL, PsycINFO, Global Index Medicus, Cochrane, and Web of Science Core Collection). Citations for all input data are provided as part of this manuscript. Study characteristics and all included data points

are provided in the Supplementary Information (Supplementary Table S1, S28). Details on data sources can also be found on the GHDx website (https://ghdx.healthdata.org/record/ihme-data/gbv-health-effects-bop-risk-outcome-scores).

# Research involving human participants, their data, or biological material

Policy information about studies with human participants or human data. See also policy information about sex, gender (identity/presentation), and sexual orientation and race, ethnicity and racism.

| Reporting on sex and gender | No primary data collection was carried out for this analysis, so the study does not involve human research participants. As stated in the methods overview, our estimates reflect men and women, drawing upon all available data regardless of how or if the input study collected and reported data by sex or gender. When we had available data, we performed sensitivity analysis by gender. |
|---|---|
| Reporting on race, ethnicity, or other socially relevant groupings | *Please specify the socially constructed or socially relevant categorization variable(s) used in your manuscript and explain why they were used. Please note that such variables should not be used as proxies for other socially constructed/relevant variables (for example, race or ethnicity should not be used as a proxy for socioeconomic status).*<br>*Provide clear definitions of the relevant terms used, how they were provided (by the participants/respondents, the researchers, or third parties), and the method(s) used to classify people into the different categories (e.g. self-report, census or administrative data, social media data, etc.)*<br>*Please provide details about how you controlled for confounding variables in your analyses.* |
| Population characteristics | No primary data collection was carried out for this analysis, so the study does not involve human research participants. The analysis evaluated the effect of physical, sexual, and psychological violence against women, and intimate partner violence and sexual violence against men on selected disease endpoints. We accepted studies regardless of the target population and exposure to violence that occurred ever, in the last three years, the past year, the past six months, the past three months, and during pregnancy in a few instances. For physical, sexual, and psychological violence, study samples represented women populations. For intimate partner violence and sexual violence, study samples represented men populations. |
| Recruitment | No primary data collection was carried out for this analysis, so we did not recruit participants. |
| Ethics oversight | This study was approved by the University of Washington IRB Committee (study #9060). |

Note that full information on the approval of the study protocol must also be provided in the manuscript.

# Field-specific reporting

Please select the one below that is the best fit for your research. If you are not sure, read the appropriate sections before making your selection.

☒ Life sciences     ☐ Behavioural & social sciences     ☐ Ecological, evolutionary & environmental sciences

For a reference copy of the document with all sections, see nature.com/documents/nr-reporting-summary-flat.pdf

# Life sciences study design

All studies must disclose on these points even when the disclosure is negative.

| Sample size | This investigation relied on existing published data. No statistical method was used to predetermine sample sizes. The number of studies included was determined through a systematic literature review that included title/abstract screening, full-text screening, and citation searching to identify relevant articles and extract data points used as input to models. Details surrounding the sample size of each included study can be found in Supplementary Table Sl and the number of included studies per risk-outcome pair is reported in Table 2. |
|---|---|
| Data exclusions | As described in Supplementary Information Section 4.2, reports were excluded based on the following exclusion criteria:<br>Study design: Cross-sectional, ecological, case series or case studies.<br>Participants: Studies conducted in subgroups identified only by convenience sampling or subgroups identified via a shared characteristic that is likely related to risk of exposure to violence or the reported health outcome (e.g., domestic violence shelter residents).<br>Exposure measurement: Studies that report only an aggregate measure of exposure combining exposure to a form of violence with other, non-eligible exposures (e.g., reports a composite ACE score only) will be excluded. For these studies, we are unable to disentangle the effect of violence exposure from the effects of other hardships or exposure types, preventing their inclusion in our review.<br>Does not meet minimum reporting criteria: Studies missing essential data, that is, those that do not report effect sizes and uncertainty information (confidence intervals, sample sizes) or the data needed to impute an effect size with uncertainty information. |
| Replication | This is a meta-analysis of existing studies with many years of cohort and other data. The code and data used are publicly available, and the analyses could theoretically be replicated. |
| Randomization | This analysis is a meta-analysis of existing studies and thus, there were no experimental groups. |
| Blinding | N/A. This study was a meta-analyses using existing data and did not the collection of primary data nor experimental/control groups. As such, blinding was not relevant. |

# Reporting for specific materials, systems and methods

We require information from authors about some types of materials, experimental systems and methods used in many studies. Here, indicate whether each material, system or method listed is relevant to your study. If you are not sure if a list item applies to your research, read the appropriate section before selecting a response.

## Materials & experimental systems

| n/a | Involved in the study |
|-----|----------------------|
| ☒ ☐ | Antibodies |
| ☒ ☐ | Eukaryotic cell lines |
| ☒ ☐ | Palaeontology and archaeology |
| ☒ ☐ | Animals and other organisms |
| ☒ ☐ | Clinical data |
| ☒ ☐ | Dual use research of concern |
| ☒ ☐ | Plants |

## Methods

| n/a | Involved in the study |
|-----|----------------------|
| ☒ ☐ | ChIP-seq |
| ☒ ☐ | Flow cytometry |
| ☒ ☐ | MRI-based neuroimaging |

## Plants

Seed stocks
: *Report on the source of all seed stocks or other plant material used. If applicable, state the seed stock centre and catalogue number. If plant specimens were collected from the field, describe the collection location, date and sampling procedures.*

Novel plant genotypes
: *Describe the methods by which all novel plant genotypes were produced. This includes those generated by transgenic approaches, gene editing, chemical/radiation-based mutagenesis and hybridization. For transgenic lines, describe the transformation method, the number of independent lines analyzed and the generation upon which experiments were performed. For gene-edited lines, describe the editor used, the endogenous sequence targeted for editing, the targeting guide RNA sequence (if applicable) and how the editor was applied.*

Authentication
: *Describe any authentication procedures for each seed stock used or novel genotype generated. Describe any experiments used to assess the effect of a mutation and, where applicable, how potential secondary effects (e.g. second site T-DNA insertions, mosiacism, off-target gene editing) were examined.*

