## [Peer Review File · Nature Human Behaviour]

The health effects associated with physical, sexual and psychological gender-based violence against men and women: a Burden of Proof study

Corresponding Author: Professor Emmanuela Gakidou

Version 0:

Decision Letter:

23rd August 2024

Dear Professor Gakidou,

Thank you once again for your manuscript, entitled "Health effects associated with exposure to sexual, physical, and psychological violence during adulthood: a Burden of Proof study." Please accept my apologies once again for the extraordinary delay in communicating a decision on your manuscript and thank you for your patience.

Your manuscript has now been evaluated by 3 reviewers, whose comments are included at the end of this letter. Although the reviewers find your work to be of interest, they also raise some important concerns. We are interested in the possibility of publishing your study in Nature Human Behaviour, but would like to consider your response to these concerns in the form of a revised manuscript before we make a decision on publication.

As you will see, the reviewers make several comments on the justification of the study, its scope and its contribution, including in light of your own previous work (reference 9). In revision, we ask that you carefully consider and address all comments. We would specifically ask that you pay particular attention to the feedback provided by Reviewer 3 -- expanding the scope of your search and outcomes would substantially strengthen the potential impact of this work and differentiate it from Spencer et al., 2023. Additionally, please do make sure to clarify the specific implications of the work in the context of the Burden of Proof methodology.

In sum, we invite you to revise your manuscript taking into account all reviewer and editor comments. We are committed to providing a fair and constructive peer-review process. Do not hesitate to contact us if there are specific requests from the reviewers that you believe are technically impossible or unlikely to yield a meaningful outcome.

We hope to receive your revised manuscript within four months. I would be grateful if you could contact us as soon as possible if you foresee difficulties with meeting this target resubmission date.

- Include a "Response to the editors and reviewers" document detailing, point-by-point, how you addressed each editor and referee comment. If no action was taken to address a point, you must provide a compelling argument. When formatting this document, please respond to each reviewer comment individually, including the full text of the reviewer comment verbatim followed by your response to the individual point. This response will be used by the editors to evaluate your revision and sent back to the reviewers along with the revised manuscript.
- Highlight all changes made to your manuscript or provide us with a version that tracks changes.

Link Redacted

We look forward to seeing the revised manuscript and thank you for the opportunity to review your work. Please do not hesitate to contact me if you have any questions or would like to discuss these revisions further.

Sincerely,

[REDACTED]

[REDACTED] PhD

Nature Human Behaviour

Reviewer expertise:

Reviewer #1: health effects of exposure to violence; measurement of violence

Reviewer #2: intimate partner violence; evidence synthesis

Reviewer #3: violence against women; gender and health; systematic review and meta-analysis

REVIEWER COMMENTS:

Reviewer #1:

Remarks to the Author:

Key results

This study reports on a meta-analysis of the health impacts of violence experienced as an adult. Novel aspects of this analysis include its focus on violence in adulthood, inclusion of findings about men (although breadth of exposure was narrower for men than women), and overall effect sizes.

Validity

The data seem to have been presented according to the methodology. Aside from some lack of clarity that is mentioned below, the results seem straightforward. The challenge is in the discussion section which does not seem to focus on the implications of the findings from this study or do not seem accurate. For example, in lines 273-274, the text suggests that maternal abortion and miscarriage and infectious diseases were revealed to be associated with violence. It may simply be a lack of textual clarity, but these issues are already known. There is also a fair number of large, mostly generic "ramifications" of the findings that do not apply specifically to this study (lines 275-319). Lines 320 to 323 seem more closely related, but still rather generically stated to know how the findings of this study relate specifically to that discussion. Instead of this more generic text, it would be helpful to focus more on the implications of this specific study.

The implication of aspects of the study's methodology would be more helpful such as the exclusion of certain violence measures and inclusion of findings where one type of violence has been adjusted for on the interpretation. What are the implications of examining the health impact of violence measured dichotomously, which may mean something very different for psychological violence than for sexual violence? What can be said for the lack of association between psychological violence and the health outcomes relative to existing literature, and within the confines of modeling psychological violence as a dichotomous variable? What do the findings about men mean for the field and how do they relate to existing literature? These types of implications, along with what can be done now with these estimates that could not have been done before without them would be very useful information.

Significance

I think the study has potential significance within its limitations, but this information was broadly missing from the study.

Data and methodology and analytic approach

The methodology is very strong and includes a large range of data with attention to temporality between exposure and outcome. Clarity issues with the lack of explanation of terms used in the results section are discussed below. The main issue with the methodology is the lack of discussion of the implication of decisions regarding inclusion and exclusion of studies based on the way violence was modeled and the more limited scope of the analysis for men.

Suggested improvements

Suggestions for improvement are included throughout the various sections of this review. The larger issues pertain to the lack of clarity in the significance of the study and mismatch between much of the Discussion section and the aspects of the analysis that are unique to this study.

Clarity and context

There is a lack of clarity in a couple of areas. The first issue is a lack of clarity due to missing details about the methodology that would allow one to better follow the study and results. When the burden of proof methodology is first mentioned, (line 96), it is entirely unclear what it is. A sentence or two explaining what this methodology is needed to better understand the results. What is the definition of an adult? What is violence perpetrated by anyone? This seems like a constructed category as the violence would have had to have been perpetrated by someone but its utility is not clear in the results section. Location is too generically used to have much meaning. At times the word is used very generically and at other times, specific locations are mentioned. Also, the

covariate adjustment and trimming jump out at the reader. Further, how is one to interpret the ROS? I understand that some of this lack of clarity has to do with the methodology being located later in the manuscript, but some interpretation of these details is needed to understand what is being reported in the results.

The other issue with clarity is in the justification for the study. There are many studies on the health effects of violence, especially intimate partner violence. There are fewer that are based on longitudinal data, this was not mentioned, simply that the health consequences of violence remain understudied, which without clarification and greater nuance about what aspects of this relationship remain understudied, is not true. There are novel aspects to the study (fewer studies including men, including violence in adulthood more comprehensively), but the implication or utility of these analyses was not made clear. What could one do with the estimates that they cannot do with existing findings?

The abstract was a bit confusing as the starting point for the study. In line 49, the wording is somewhat confusing “we systematically review and meta-analysis to evaluate”. Also, the number of health outcomes per violence of violence exposure would be helpful to know. Seven health outcomes were associated with exposure to physical violence...how many were not? Word limits likely preclude as many details as would be desired, but with the current results, it is not clear how more data and research and services are clear next steps.

References

The references seem to have been reported per conventions.

Reviewer #2:

Remarks to the Author:

It is my privilege to review the manuscript. In alignment with the high research standards of the IHME, the large-scale systematic review and meta-analysis provided significant insights into the health impacts of physical, sexual, and psychological violence against women, as well as intimate partner violence and sexual violence against men—a topic that has received less attention in health research and practice. The study contributed to both the Burden of Proof research series, and the Lancet Commission on Gender-based Violence and Maltreatment of Young People, offering synthesized evidence to address the critical, yet preventable, public health issue of interpersonal violence. The findings will make a meaningful contribution to the policy-making and health system’s response to violence prevention. As an IPV researcher, I commend the IHME team for their rigorous methodology and substantial contributions to this field. My comments are minor.

Page 3, Line 64 to 65: The term “violence” in the first sentence was overly broad and vague. Additionally, the statement “more than one billion people” was inconsistent with the cited sources, as this figure pertains to violence against children (<https://www.who.int/news-room/fact-sheets/detail/violence-against-children>).

Please revise the sentence to specify the type of violence and the number of people affected in the context of the study, and provide appropriate references.

Page 3, Line 66: The phrase “...sexual, physical, and psychological violence...” appeared in an inconsistent order. I recommend using “physical, sexual, and psychological violence” consistently throughout the manuscript, as it was more commonly used elsewhere in the text.

Page 4, Line 78 to 79: The sentence “...physical, sexual, and psychological violence against women and IPV and sexual violence against women and men” should be revised to “...physical, sexual, and psychological violence against women, and IPV and sexual violence against men” for clarity.

Page 4, Line 91, and Page 19, Line 427:

(1) The inception date of “January 1, 1970” should be briefly justified, at least in the response letter if not in the manuscript.

(2) Please ensure consistency in date formatting. The manuscript used both “January 1, 1970” and “1 January 1970” (Page 19, Line 427).

(3) There is a discrepancy in the end dates for the studies included. Page 4 mentioned “January 31, 2023,” while Page 19 mentioned “January 31, 2021.” Additionally, the 2022 protocol published in BMJ Open specified an end date of September 30, 2021. Updated searches are common and recommended, and the authors clearly stated that an updated search was conducted in the Supplementary Information Section 4. Please correct the end date as “January 31, 2023,” where relevant throughout the manuscript.

Page 5, Line 114, Page 6, Line 130: The term “sexually transmitted infections” first appeared on Page 5, and “sexually transmitted diseases (STIs)” on Page 9, Line 208. Please introduce the abbreviation “STI” here (rather than on Page 6) and use it consistently throughout the manuscript. Also, ensure consistent usage of terms like “excluding HIV/AIDS” across the text.

Page 7, Line 141: Please provide the citation for “one observation which measured IPV.”

Page 11, Line 237: The sentence “...psychological violence exposures included any psychological violence against women but for men, ...” would benefit from a comma before “but for men” to improve readability.

Page 11, Line 241 to Line 242: Please revise “self-harm (number of included studies (n) = 3” to “self-harm (n = 3).”

Page 12, Line 260 to 261: The sentence “(See Table 2 for estimated RRs and accompanying UIs, both excluding and including gamma [between-study heterogeneity], with the star rating based on the latter.)” was somewhat unclear. Please consider rephrasing for clarity.

Page 13, Line 288 to Line 295, Line 296 to Line 306: The paragraphs effectively summarized the challenges faced by survivors and the health system's response. However, the references (No. 94 to No. 96) in the paragraph seemed to focus more on women and abortion, which may not fully align with the broader topic discussed. Moreover, while the Brave Movement (No. 101) is significant, it is less relevant to this study which focused on adults. Additionally, barriers within the health system also impede healthcare providers from identifying and managing IPV, child abuse, and other forms of interpersonal violence. To enhance the argument and encompass a wider perspective (for example, including IPV against men), I suggest incorporating the following articles and adding one to two sentences describing the complexity of the issue at different levels:

Taylor JC, Bates EA, Colosi A, Creer AJ. Barriers to Men's Help Seeking for Intimate Partner Violence. *J Interpers Violence*. 2022 Oct;37(19-20):NP18417-NP18444.
(IPV against men)

Heron RL, Eisma MC. Barriers and facilitators of disclosing domestic violence to the healthcare service: A systematic review of qualitative research. *Health Soc Care Community*. 2021 May;29(3):612-630.
(Healthcare service)

Li Q, Zeng J, Zhao B, Perrin N, Wenzel J, Liu F, Pang D, Liu H, Hu X, Li X, Wang Y, Davidson PM, Shi L, Campbell JC. Nurses' preparedness, opinions, barriers, and facilitators in responding to intimate partner violence: A mixed-methods study. *J Nurs Scholarsh*. 2024 Jan;56(1):174-190.
(Nurses' response in low-resource locations)

Page 15, Line 342: Please provide the link for the website recourse (No. 107) and the website link for reference No. 3.

Page 18 to Page 19, Line 423 to 425: The sentence "...a study already reported results specifically to exposure to physical or sexual intimate partner violence against women and child sexual abuse⁹" should be revised based on the following points:
(1) Citation No.9, this 2023 study published in *Nature Medicine* ("Health effects associated with exposure to intimate partner violence against women and childhood sexual abuse: a Burden of Proof study"), also included psychological IPV against women. Please confirm with the IHME team, and revise the sentence accordingly.
(2) The 2023 *Nature Medicine* study included 57 articles on IPV against women, some of which were also included in the current study (eg, Leung 2002 from Hong Kong, Ibrahim 2015 from Egypt). Is there any overlap in the findings between this article and the current study? If so, what is the proportion of overlap? If not, please clarify this briefly in the manuscript.
(3) How does the current study supplement the 2023 *Nature Medicine* study and the Lancet Commission on Gender-based Violence regarding the included types of violence against women? Given the differing terms and definitions used in IPV research, any vague expressions could lead to confusion. Please provide an additional sentence clarifying these points, possibly on Page 19, Line 425.

Page 19, Line 437: Please add the closing parenthesis ")" to the sentence.

Page 19, Line 442: Please delete the word "is" in the phrase "including but is not limited to."

Page 20, Line 469: The term "gender-based violence" was first mentioned on Page 4 Line 80. Please introduce the abbreviation "GBV" on Page 4 and use it consistently throughout the manuscript. Check for consistency with other terms as well.

Page 22, Line 507: Please correct the word "selected" by using the past tense.

Page 22, Line 511: The manuscript contained inconsistent use of hyphen (-), shorter en dash (–), and longer em dash (—). For example, in "Supplementary Tables S18–S19," the shorter en dash should be used, not the hyphen (-). Please correct these issues throughout the manuscript and supplementary materials. Although time-consuming, these revisions are essential, especially for an article from the IHME.

Page 26, Line 591: Please insert a space between "In" and "(BPRF)."

Supplementary information Figure S1: Please re-create the screenshot of the PRISMA diagram to remove unnecessary error notifications, such as those under "screened," "excluded," and "retrieval."

Reviewer #3:

Remarks to the Author:

This study estimates the health-related impacts of exposure to multiple forms of violence in adulthood. It expands upon existing efforts to measure the health impacts of exposure to violence against women by their male intimate partners by including physical, sexual, or psychological violence perpetrated by anyone against adult women and IPV or sexual violence against adult men. As such it makes an important, new contribution to the on-going effort to establish exposure to violence as a significant risk factor contributing to the global burden of disease. The findings should be of immediate interest to the fields of public health, victimology, criminology, social work, as well as those particularly interested in violence against women and other forms of gender-based abuse.

The scope of the review, however, could be better justified and more consistently expressed. The abstract initially claims, for example, to examine the health impacts of violence—which could suggest that among the exposures examined would be things like community or gang violence and simple assault. The abstract goes on to clarify that the focus is on violence by all perpetrators against women, and IPV or sexual violence against men. Later in the introduction the authors describe their work as: "the most extensive systematic review on the health effects associated with exposure to violence." Similar slippages to "health effects of

violence” happen throughout the manuscript.

Moreover, the authors do not offer a clear explanation for the types of violence they have chosen to include. One could ask why include physical violence against women by acquaintances or strangers but not for men? Nor does the search strategy suggest that the types of physical assault that might be captured in crime data have been consciously included for either men or women. This leads me to believe that the goal is not to establish the full health effects of victimization as a risk factor for health outcomes, but to update the existing risk factor analysis on violence against women by adding data on IPV and sexual assault against men. In another place, the analysis is framed as analysis of the health impact of violence directed at specific groups of adults, presumably because of their group identity. If this were the underlying rationale, one would expect search terms for hate crimes and physical and sexual violence against sexual and gender minorities to be included.

Presumably, the framing of the review (and hence types of studies included), is either a function of the fact that the analysis is derived from a larger study on violence against women or because the review is intended to focus on gender-based violence only—that is types of violence that are influenced, at least in part, by gender-related norms and power relations. It would be useful for the authors to describe what led them to their decisions on which types of violence to include (in terms of the author's policy intent, paradigm, or circumstance, e.g., membership on the Lancet Commission on GBV and Maltreatment of Children).

Data & methodology: The manuscript is a systematic review and meta-analysis of existing literature that assesses the potential incremental risks posed by exposure to physical, sexual, or psychological violence in adulthood. The methods for the overarching systematic review and meta-analysis are well documented and well-executed.

I am unable to evaluate the underlying Burden of Proof Risk Function methodology, because the methods exceed my quantitative skills. I do have several suggestions and questions, however, regarding other elements of the methodology.

1. Selection of Studies: As I was reading and checking references, I repeatedly thought of studies that I would have expected to see included that were not. For example:

Kate Doyle, Ruti G. Levto, Emmanuel Karamage, et al. Long-term impacts of the Bandebereho programme on violence against women and children, maternal health seeking, and couple relations in Rwanda: a six-year follow-up of a randomised controlled trial. *eClinicalMedicine* 2023;64: 102233 Published Online 26 September 2023 <https://doi.org/10.1016/j.eclinm.2023.102233>

Chatterji, S., Heise, L., 2021. Examining the bi-directional relationship between intimate partner violence and depression: findings from a longitudinal study among women and men in rural Rwanda. *SSM-Mental Health* 1, 100038.

T. Muhammad, Saddaf Naaz Akhtar, Waad Ali, Chanda Maurya, Cross-lagged relationships between exposure to intimate partner violence, depressive symptoms and suicidal thoughts among adolescent and young married women, *Journal of Affective Disorders*, Volume 360, 2024, Pages 259-267, ISSN 0165-0327, <https://doi.org/10.1016/j.jad.2024.05.088>.

Devries, K.M., Mak, J.Y., Bacchus, L.J., et al., 2013. Intimate partner violence and incident depressive symptoms and suicide attempts: a systematic review of longitudinal studies. *PLoS Med.* 10 <https://doi.org/10.1371/journal.pmed.1001439>.

Part of the problem is that the discussion of methods comes at the end of the article, which leaves the reader wondering the boundaries of inclusion. Even in the delayed methods section, nowhere does it clearly state the inclusion and exclusion criteria for studies. It wasn't until I examined the supplementary material that I came to realize what types of studies (cohort, case control, etc) and what type of data (administrative, survey, police reports, etc.) qualified for establishing a risk association. Also, very important from the perspective of the violence against women field, is the decision to exclude studies that use combined outcome measures like physical and/or sexual violence by an intimate partner. I would suggest moving a combined version of Tables S13 and S14 to the methods section or at least clarifying early in the manuscript the types of studies and outcomes that qualify for inclusion in the review.

2. Outcome inclusion. With respect to IPV, I find the decision to exclude studies with a composite outcome measure from the review, problematic. Historically, the standard measure for studies on IPV has been physical and/or sexual violence by a current and/or former partner. This is because most women experience IPV as a pattern of different types of violence and controlling behaviors over time. When creating a “case” of violence in the context of prevalence or risk factor studies, the traditional physical and/or sexual violence measure captured the mix experiences that women report who meet the definition of IPV. Most of those who experience sexual IPV or physical IPV also experience emotional abuse and episodes that include amalgams of different types. I can see the benefits of exploring the different types of violence separately, but I suspect that if you additionally had allowed for composite measures for IPV, you would have had more studies qualifying for inclusion.

I also think you would have a wider evidence base if you had not restricted studies to only women over 18; there are a number of well-known prospective cohorts of young people being followed (e.g. Ecuador, Peru, Vietnam, India) overtime that collect information on IPV and health outcomes. Many of these follow adolescents into their adult years (such as the Young Lives Study out the University of Oxford or the UDAYA study cited above).

Additionally, it seems that you may be missing relevant data on IPV and health outcomes from prospective cohorts embedded in impact evaluations/RCTs. Both the Doyle and Chatterji analysis come from evaluation of IPV prevention interventions in Rwanda.

Obviously, it is the authors prerogative to establish the inclusion and exclusion criteria, but I suspect some of the decisions made have needlessly reduced the data available for inclusion in the systematic review.

3. Outcome construction. One additional thing I did not see addressed is whether or not different investigators used a “clean” reference group when constructing their IPV exposure variable. Since individuals experiencing IPV so often experience multiple

forms of violence during any episode or over time, binary variables based on "have experienced" versus "have not experienced" tend to underestimate the strength of association unless a special effort is made to remove women who have experienced any other type of IPV from the reference group. If women experienced a combination of physical and sexual violence from the same partner and she is coded "yes" = 1 for physical violence, and everyone else becomes zero, then women experiencing other types of IPV will be in the reference group, thus "diluting" the strength of association between physical violence and IPV. There is a mix of practice in the field regarding "clean" versus "non-clean" reference groups, so you may want to add this as a bias test in your meta-analysis.

This is discussed in the Devries et al and Chatterji et al articles above and in this measurement brief:

Overall, the manuscript is an important contribution to the literature. It would make sense in the future to do a similar analysis of the impact of any victimization in childhood, adolescence and adulthood on negative health outcomes and to explore if the risk of negative outcomes increases as the number and types of violence experiences likewise increase.

Version 1:

Decision Letter:

Our ref: NATHUMBEHAV-24020709A

11th December 2024

Dear Dr. Gakidou,

Thank you for submitting your revised manuscript "Health effects associated with exposure to sexual, physical, and psychological violence during adulthood: a Burden of Proof study" (NATHUMBEHAV-24020709A). It has now been seen by Reviewers 2 and 3 from the previous round of review and their comments are below. As you can see, the reviewers find that the paper has improved in revision. We will therefore be happy in principle to publish it in Nature Human Behaviour, pending minor revisions to comply with our editorial and formatting guidelines.

We are now performing detailed checks on your paper and will send you a checklist detailing our editorial and formatting requirements within two weeks. Please do not upload the final materials and make any revisions until you receive this additional information from us.

Sincerely,

[Redacted]

[Redacted] PhD

[Redacted]
Nature Human Behaviour

Reviewer #2 (Remarks to the Author):

I am satisfied with the revised version of the manuscript. All of the comments have been thoroughly and appropriately addressed.

Reviewer #3 (Remarks to the Author):

I have fully reviewed the revised manuscript and appendices and feel that the authors have adequately addressed the reviewers concerns. I would support publication of this version without further revisions.

Version 2:

Decision Letter:

Dear Emmanuela,

I am very pleased to inform you that your Article "The health effects associated with physical, sexual and psychological gender-based violence against men and women: a Burden of Proof study", has now been accepted for publication in Nature Human Behaviour.

Authors may need to take specific actions to achieve <https://www.springernature.com/gp/open-research/funding/policy-compliance-faqs> compliance with funder and institutional open access mandates. If your research is supported by a funder that requires immediate open access (e.g. according to <https://www.springernature.com/gp/open-research/plan-s-compliance> Plan S principles) then you should select the gold OA route, and we will direct you to the compliant route where possible. For authors selecting the subscription publication route, the journal's standard licensing terms will need to be accepted, including <https://www.springernature.com/gp/open-research/policies/journal-policies> self-archiving policies. Those licensing terms will supersede any other terms that the author or any third party may assert apply to any version of the manuscript.

With warm regards,

[Redacted]

[Redacted] PhD

[Redacted]
Nature Human Behaviour

P.S. Click on the following link if you would like to recommend Nature Human Behaviour to your librarian
<http://www.nature.com/subscriptions/recommend.html#forms>

** Visit the Springer Nature Editorial and Publishing website at http://editorial-jobs.springernature.com?utm_source=ejp_NHumB_email&utm_medium=ejp_NHumB_email&utm_campaign=ejp_NHumB for more information about our career opportunities. If you have any questions please click [here](mailto:editorial.publishing.jobs@springernature.com).

Open Access This Peer Review File is licensed under a Creative Commons Attribution 4.0 International License, which permits use, sharing, adaptation, distribution and reproduction in any medium or format, as long as you give appropriate credit to the original author(s) and the source, provide a link to the Creative Commons license, and indicate if changes were made. In cases where reviewers are anonymous, credit should be given to 'Anonymous Referee' and the source. The images or other third party material in this Peer Review File are included in the article's Creative Commons license, unless indicated otherwise in a credit line to the material. If material is not included in the article's Creative Commons license and your intended use is not permitted by statutory regulation or exceeds the permitted use, you will need to obtain permission directly from the copyright holder.

Response to Reviewers for NATHUMBEHAV-24020709:

Physical, sexual, and psychological violence: A Burden of Proof study on the health effects associated with gender-based violence against men and women

REVIEWER COMMENTS:

Reviewer #1:

Key results

1. This study reports on a meta-analysis of the health impacts of violence experienced as an adult. Novel aspects of this analysis include its focus on violence in adulthood, inclusion of findings about men (although breadth of exposure was narrower for men than women), and overall effect sizes.

We appreciate the time the reviewer took to review our manuscript, and also their recognition of the contributions our work makes to this field of research.

Validity

2. The data seem to have been presented according to the methodology. Aside from some lack of clarity that is mentioned below, the results seem straightforward. The challenge is in the Discussion section which does not seem to focus on the implications of the findings from this study or do not seem accurate. For example, in **lines 273-274**, the text suggests that maternal abortion and miscarriage and infectious diseases were revealed to be associated with violence. It may simply be a lack of textual clarity, but these issues are already known. There is also a fair number of large, mostly generic “ramifications” of the findings that do not apply specifically to this study (**lines 275-319**). **Lines 320 to 323** seem more closely related, but still rather generically stated to know how the findings of this study relate specifically to that discussion. Instead of this more generic text, it would be helpful to focus more on the implications of this specific study.

Thank you for highlighting this concern. We have made significant revisions to the Discussion, reframing the text to focus on the implications of our study more specifically. Following the suggestion from the reviewer, we now more clearly note that our results confirm prior findings about the relationship between exposure to GBV and poor mental health and extend the focus beyond mental health to other conditions such as maternal and reproductive health. In addition, we use the opportunity to further clarify the unique contribution of the Burden of Proof methodology to the interpretation of the findings:

Lines 279-285: “Overall, our analysis confirmed past findings about the relationship between exposure to GBV and poor mental health^{7–10} while highlighting the fact that the health effects of GBV extend beyond mental health, including substance abuse disorders and maternal and reproductive health. Overwhelmingly, however, the consistently weak evidence of associations between GBV and health outcomes identified by our methods that uniquely incorporate between-study heterogeneity into results illuminates the need for more research to further our understanding of known health associations to GBV and to extend the evidence on GBV’s health consequences beyond IPV against women.”

Additionally, within the updated Discussion, new focus has been placed on a few key topics including: 1) how the existing data landscape systematically overlooks gender-based violence victimization as a health risk factor for men, 2) the paucity of data on the health effects of psychological GBV compared to data on physical or sexual GBV, 3) implications of the variability in the underlying data and overall low number of included studies, and 4) the crucial need to bolster both prevention programming and health services for survivors. With these adjustments, we have removed the generic language and broad focus to provide a more nuanced discussion that better emphasizes the important implications of this work.

3. The implication of aspects of the study's methodology would be more helpful such as the exclusion of certain violence measures and inclusion of findings where one type of violence has been adjusted for on the interpretation. What are the implications of examining the health impact of violence measured dichotomously, which may mean something very different for psychological violence than for sexual violence? What can be said for the lack of association between psychological violence and the health outcomes relative to existing literature, and within the confines of modeling psychological violence as a dichotomous variable? What do the findings about men mean for the field and how do they relate to existing literature? These types of implications, along with what can be done now with these estimates that could not have been done before without them would be very useful information.

We have made significant revisions to the Discussion and limitations to include further detail related to the implications of the study's methodology.

1. We expanded the limitations to better discuss the reasons for and implications of using dichotomous measures of violence exposure (lines 387-398) and included further language exploring co-occurrence of violence in lines 367–386. Specifically, while we recognize that it is crucial to consider co-occurrence when evaluating the health effects of gender-based violence, the landscape of existing literature restricts our ability to do so in the present analysis. Many of the studies we identified did not explicitly state their inclusion or exclusion of individuals with exposures to other forms of violence and, among those explicitly examining combined forms of violence, there were inconsistencies in the combinations being explored. Consequently, any adjustment to account for co-occurrence would be reliant on substantial and unfounded assumptions on the research team's part. However, we believe that our approach brings to light the important and substantial health consequences of these individual forms of violence while acknowledging that our estimates represent the minimum additional risk experienced by a survivor in the context of co-occurrence likely compounding health risks.

Additionally, in the revised text, we explain why it is not feasible to look at dose-response associations given the existing evidence and acknowledge that this is an important area for future research. The goal of this work is explicitly to both assess the current landscape of evidence and identify comparable and concrete areas where new evidence is necessary. In the future, our goal is to move towards a dose-response model, but until the evidence we would need to generate this type of model is available, all we can do is include it in our call to action.

Lines 387-398: *“A further limitation is the use of dichotomous case definitions for physical, sexual, and psychological GBV. This approach does not allow consideration of how differences in frequency or severity may interact with the health consequences of exposure*

to GBV. However, related to the research gaps discussed above, few of the studies we included reported effect sizes for severity- or frequency-stratified exposed groups relative to an unexposed group. Where this information was available, the categories were varied, preventing meaningful comparison. Furthermore, as exemplified by the paucity of data on male-specific observations in addition to an over-representation of data from high-income countries, included studies largely reflected similar populations of women in high-income countries and did not provide further layers of granularity to the data (see Supplemental Table S1 for relevant metadata). Given that different sub-populations may experience different forms of violence at different rates or severity levels^{76,78,79}, future research on the frequency- and severity-related health effects of GBV exposure in adulthood and among sub-groups, particularly non-binary individuals, is crucial.”

2. We have added a section in the discussion to more thoroughly discuss our findings on psychological violence and the lack of association between psychological violence and the health outcomes relative to existing literature. Our discussion now highlights important discrepancies that may result in underestimation of the risks associated with psychological violence—even more so than for other forms of GBV, including that psychological GBV is particularly overlooked in data collection, there is substantial inconsistency in the definition of psychological violence, and there is a lack of consensus on what qualifies as psychological GBV, globally.

Lines 303-309: “In addition to the women-specific focus of existing literature, there is a clear lack of data on the health effects of psychological GBV compared to data on physical or sexual GBV. While any form of GBV is subject to under-reporting and under-detection⁵⁸, psychological GBV is particularly overlooked, with substantial inconsistency in the definition and recognition of this form of violence⁵⁹. Additionally, while many of the studies identified in our present analysis use self-reported exposure to GBV to identify survivors, psychological violence manifests in ways that may be challenging for the survivors themselves to identify⁶⁰ and has been linked to feelings of shame and stigma^{61–64}.”

3. Moreover, we now better discuss our findings about men, and how they relate to the broader field of violence in lines 285-302. We highlight, for example, how our results align with existing literature pointing to the significant gap in the existing data landscape, which systematically overlooks GBV victimization as a health risk factor for men:

Lines 292-302: “[...] Despite their inclusion, these data points are still minimal compared to the number of women-specific data points available, reflecting an imbalance in the existing data landscape that systematically overlooks GBV victimization as a health risk factor for men⁵⁵. However, the consequences of GBV—regardless of the gender of the victim—are demonstrated by our present results, even in the context of our highly conservative interpretation of the evidence available. In lieu of a broader evidence base for men to allow for further granularity, we have to assume that there are no differences in the health consequences of GBV victimization between men and women, despite broad consensus of the gendered characteristics of the associated health outcomes^{5,56,57} that make this assumption unlikely and deeply flawed. To begin understanding and responding to these possible differences, GBV perpetrated against men must not be overlooked in future research, policy, or programming.”

4. Finally, with the extension revision of our manuscript, we further clarify the contribution of our to the field, as suggested by the reviewer (lines 76-87). Most existing systematic reviews focus either on the broad exposure of sexual violence or IPV, and typically restrict their search to a single health outcome or focus on a group of health outcomes, but restrict to only studies published in English. Our review is distinct because 1) it allows for direct comparisons between different policy and programmatically relevant subtypes of GBV, 2) it includes findings for any health outcome, which allows us to capture the full breadth of health consequences of GBV, and 3) it is not limited by publication language or location. Additionally, as noted above, our work stands out from previous meta-analyses by incorporating effect sizes for GBV against men into our estimates of adverse health outcomes associated with physical, sexual, and psychological GBV, which is a critically important step to better understand and respond to GBV perpetrated against men. Finally, in our comprehensive synthesis of available evidence without constraining our focus to specific kinds of health outcomes, we directly inform the on-going effort to establish exposure to violence as a significant risk factor contributing to global disease burden. As a result of this approach, we have identified all health outcomes with sufficient data that should be associated with GBV among adults in the GBD study, which will allow for robust estimates of disease burden due to GBV directly comparable to the burden due to other risk factors.

Significance

4. I think the study has potential significance within its limitations, but this information was broadly missing from the study.

We have significantly revised the language used throughout the Introduction, Results, Discussion, and Methods of the manuscript to clarify both the significance and contribution of the study. In all sections, we now explicitly differentiate the scope of our analysis from that of previously published work and the existing evidence base.

Lines 92-100 “This work builds off a prior effort to quantify and evaluate the health effects of IPV and Childhood Sexual Abuse^{12,13} by taking a broader lens on GBV exposure and its associated health consequences. We extended our analysis of GBV exposure to include physical, sexual, and psychological GBV against women by any perpetrators, and to include data on men’s exposure to physical and psychological IPV and sexual violence. Expanding our exposure scope allows us to disentangle the different manifestations of GBV to provide a more accurate representation of the complex relationships between GBV exposure and health outcomes beyond the common focus of IPV among women and, in the future, to allow for a more comprehensive estimation of the disease burden attributable to GBV.”

Lines 76-84: “Despite this progress, significant research gaps remain that hinder our ability to more effectively address the impacts of GBV. For example, not all existing evidence properly controls for the timing between exposure and outcome, with few longitudinal studies being available, limiting the evidence base for establishing causality between GBV and health outcomes. Moreover, most existing studies focus on intimate partner violence (IPV) against women and most often do not distinguish the health consequences associated with specific types of violence (e.g., the consequences of sexual, physical or psychological violence separately). Additionally, while the role of men as common perpetrators of GBV has long been recognized, their position as victims has historically been understudied, resulting in limited data availability on the health consequences for male survivors”.

We have also clarified the strengths of the Burden of Proof methodology throughout the manuscript, emphasizing how the methodology allows us to better synthesize the available evidence and quantify evidence strength. Particularly, our work showcases the importance of quantifying the strength of available evidence in a low-research environment so that we do not under or over-state associations. This work's goal is explicitly to both lay out the current landscape of evidence and identify comparable and concrete areas where new evidence is necessary. We fully revised the implications of work in Table 1.

Data and methodology and analytic approach

5. The methodology is very strong and includes a large range of data with attention to temporality between exposure and outcome. Clarity issues with the lack of explanation of terms used in the results section are discussed below. The main issue with the methodology is the lack of discussion of the implication of decisions regarding inclusion and exclusion of studies based on the way violence was modeled and the more limited scope of the analysis for men.

We have added more details on the burden of proof methods to the Introduction and the Methods section. We also extensively revised our Results section to enhance clarity around the terms used. We have further expanded upon our inclusion and exclusion criteria in the Methods (lines 476-488). As noted previously, we have also added language discussing how the lack of evidence on GBV victimization as a health risk factor for men limits our analysis (lines 285-302).

Suggested improvements

Suggestions for improvement are included throughout the various sections of this review. The larger issues pertain to the lack of clarity in the significance of the study and mismatch between much of the Discussion section and the aspects of the analysis that are unique to this study.

We appreciate the reviewer highlighting the areas in which we can improve the clarity and precision of our manuscript, and where possible, we have made changes throughout the Discussion section to provide further clarification. Please find our responses to each distinct concern below.

Clarity and context

1. There is a lack of clarity in a couple of areas. The first issue is a lack of clarity due to missing details about the methodology that would allow one to better follow the study and results. When the burden of proof methodology is first mentioned, (**line 96**), it is entirely unclear what it is. A sentence or two explaining what this methodology is needed to better understand the results. What is the definition of an adult? What is violence perpetrated by anyone? This seems like a constructed category as the violence would have had to have been perpetrated by someone but its utility is not clear in the results section. Location is too generically used to have much meaning. At times the word is used very generically and at other times, specific locations are mentioned. Also, the covariate adjustment and trimming jump out at the reader. Further, how is one to interpret the ROS? I understand that some of this lack of clarity has to do with the methodology being located later in the manuscript, but some interpretation of these details is needed to understand what is being reported in the results.

Thank you for highlighting these clarity issues. We have revised our language to provide greater clarity across the manuscript with a particular focus on the Methods and Introduction. Specific instances include:

1. We added some clarification about what the Burden of Proof methodology and how to interpret the BPRF, ROS, and star rating to the Introduction, as well as clarifying the latter terms more clearly in the Methods (lines 637-659):

Lines 103-113: “This methodology allows us to both systematically evaluate the potential association between the exposure of interest and a given health outcome and to quantify the strength of the underlying evidence. In addition to producing conventional measures of association, the Burden of Proof methodology generates conservative measures that account for both known and unknown sources of heterogeneity across input studies. The Burden of Proof Risk Function (BPRF) can be translated into both a Risk-Outcome Score (ROS) and an estimate of the minimum percent of increased health risk attributable to GBV exposure. The ROS communicates both the magnitude of the association and the strength of the underlying data, with greater positive values reflecting a larger effect size and/or stronger evidence. The ROS can in turn be converted into easily comparable and interpretable star ratings ranging from one (weak) to five (strong) that categorize significant associations according to effect size and evidence strength”.

*Lines 643-652: “[...] the BPRF incorporates both the degree of certainty in the point estimate and the underlying variation in the data. It can be conceptualized as an increase in the risk of the disease outcome by at least the value of $(BPRF - 1) * 100$ among exposed individuals and can be transformed into a Risk-Outcome Score (ROS) of $\log(BPRF)/2$. A higher positive ROS means that a relationship between exposure and outcome is characterized by a larger effect size and/or strong underlying evidence, while a lower and negative ROS indicates weak evidence of a relationship.*

To make ROS easier to interpret by policymakers and research funders and easier to compare between dichotomous and continuous risk-outcome pairs, ROS values are mapped onto a star-rating system from one to five stars based on pre-established thresholds.”

2. We removed mentions of location as we agree that the term was ambiguous and not relevant to the core findings. This was also done to reduce manuscript length.
3. We revised our language in the Introduction and in the Methods to clearly define adulthood. The focus of the study was on men and women at or above the age of 18. These language adjustments can be seen in Lines 91 and 102 in the Introduction and lines xx in the Methods section):

Lines 90-92: “[...] provides a comprehensive systematic review of the health effects associated with exposure to physical, sexual, and psychological GBV against adults (men and women aged 18 years and older).”

Lines 101-102: “[...] we estimated the associations between physical, sexual, and psychological GBV against men and women aged 18 and older...”

4. Violence perpetrated by anyone refers to a run of the analysis subset to data that did not restrict the exposure by the types of perpetrators. We have adjusted our language in line 142 from “perpetuated by anyone” to “gender-based violence from unspecified or unrestricted perpetrators.” The adjusted language was inserted throughout the rest of the manuscript where appropriate. We also adapted the language in the Introduction:

*Lines 94-99: “We extended our analysis of GBV exposure to include physical, sexual, and psychological GBV against women, **regardless of the perpetrator**, and to include data on men’s exposure to physical and psychological IPV and sexual violence. Expanding our exposure scope allows us to disentangle the different manifestations of GBV to provide a more accurate representation of the complex relationships between GBV exposure and health outcomes beyond the common focus of IPV among women and, in the future, to allow for a more comprehensive estimation of the disease burden attributable to GBV”.*

2. The other issue with clarity is in the justification for the study. There are many studies on the health effects of violence, especially intimate partner violence. There are fewer that are based on longitudinal data, this was not mentioned, simply that the health consequences of violence remain understudied, which without clarification and greater nuance about what aspects of this relationship remain understudied, is not true. There are novel aspects to the study (fewer studies including men, including violence in adulthood more comprehensively), but the implication or utility of these analyses was not made clear. What could one do with the estimates that they cannot do with existing findings?

Thank you for highlighting the need for clearer language to justify this study. We have added text in the Introduction to detail where there are gaps in the current landscape of peer-reviewed literature and directly discuss the lack of longitudinal data, limited studies on GBV among men, and the historical focus on women exposed to IPV:

Lines 76-84: “Despite this progress, significant research gaps remain that hinder our ability to more effectively address the impacts of GBV. For example, not all existing evidence properly controls for the timing between exposure and outcome, with few longitudinal studies being available, limiting the evidence base for establishing causality between GBV and health outcomes. Moreover, most existing studies focus on intimate partner violence (IPV) against women and most often do not distinguish the health consequences associated with specific types of violence (e.g., the consequences of sexual, physical or psychological violence separately). Additionally, while the role of men as common perpetrators of GBV has long been recognized, their position as victims has historically been understudied, resulting in limited data availability on the health consequences for male survivors”.

Additionally, throughout the revised version of the manuscript, we have added language in the to clarify why our findings are important and how they extend the existing literature. In contrast to most prior systematic reviews, which largely focus on exposure to sexual violence or IPV broadly, our review aims to identify and compare the health consequences of three different forms of GBV, providing policy-relevant insights into forms of GBV that require distinct interventions. Moreover, our review was not restricted to specific health outcomes, unlike most previous reviews that narrowly focus primarily on some subset of reproductive outcomes,

mental health, or substance use disorders. This broad approach allowed us to capture the wide breadth of health consequences of GBV without preconceptions of what health outcomes may be the most relevant, while drawing on the GBD framework ensured that our outcome definitions were rigorous and clinically relevant. Lastly, our findings provide robust and highly specified details about the relationships between GBV and health outcomes in a way that allow for comparisons with other risk factors in studies like the Global Burden of Disease study. This is important because what is lacking in the field is a way to compare exposure to GBV to other risk factors, such as obesity, stunting and wasting, air pollution and thus putting exposure to violence on the same scale as other priorities and highlighting its magnitude.

3. The abstract was a bit confusing as the starting point for the study. In **line 49**, the wording is somewhat confusing “we systematically review and meta-analysis to evaluate”. Also, the number of health outcomes per violence of violence exposure would be helpful to know. Seven health outcomes were associated with exposure to physical violence...how many were not? Word limits likely preclude as many details as would be desired, but with the current results, it is not clear how more data and research and services are clear next steps.

We appreciate the reviewer’s suggestion, and we have carefully revised the entire abstract to improve and simplify the language. As the reviewer noted, the abstract word limits prevent us from adding significant detail; however, we have incorporated further information on the number of health outcomes analyzed for each form of violence and the relevant Burden of Proof measures.

References

4. The references seem to have been reported per conventions.

We have confirmed that our references meet the standard format required by the Nature family of journals and have ensured any new references follow the same format.

Reviewer #2:

1. It is my privilege to review the manuscript. In alignment with the high research standards of the IHME, the large-scale systematic review and meta-analysis provided significant insights into the health impacts of physical, sexual, and psychological violence against women, as well as intimate partner violence and sexual violence against men—a topic that has received less attention in health research and practice. The study contributed to both the Burden of Proof research series, and the Lancet Commission on Gender-based Violence and Maltreatment of Young People, offering synthesized evidence to address the critical, yet preventable, public health issue of interpersonal violence. The findings will make a meaningful contribution to the policy-making and health system’s response to violence prevention. As an IPV researcher, I commend the IHME team for their rigorous methodology and substantial contributions to this field. My comments are minor.

We are thankful for this reviewer’s kind words and recognition of our efforts and methodological and thematic contributions. We set out to comprehensively evaluate existing evidence pertaining to the health consequences of these forms of violence, and we are heartened to see the reviewer’s recognition that we have achieved our goal from the perspective of an IPV researcher. Please find our responses to the reviewer’s comments and corresponding revisions described below.

2. Page 3, Line 64 to 65: The term “violence” in the first sentence was overly broad and vague. Additionally, the statement “more than one billion people” was inconsistent with the cited sources, as this figure pertains to violence against children (<https://www.who.int/news-room/fact-sheets/detail/violence-against-children>).

Please revise the sentence to specify the type of violence and the number of people affected in the context of the study, and provide appropriate references.

We appreciate the reviewer highlighting this inconsistency. We’ve removed the statement related to the number of people affected by violence in lines 62-63. We also made significant adjustments to the language in this section to better clarify the focus of our analysis. In lines 63-65 we have defined gender-based violence: “GBV is defined as violence directed at an individual based on their biological sex, gender identity, gender expression or failure to adhere to socially defined norms of masculinity and femininity³.” and adjusted our language to better clarify the types of violence analyzed in lines 88-92:

Lines 88-92: “Our study—which contributes to the goals of the Lancet Commission on Gender-based Violence and Maltreatment of Young People² to expand the consideration of GBV as an important global health concern for all people—provides a comprehensive systematic review of the health effects associated with exposure to physical, sexual, and psychological GBV against adults (men and women aged 18 years and older).”

To ensure consistency, we have adjusted our language throughout the document to better clarify our focus on gender-based physical, sexual, and psychological violence, and provided a clearer justification for how we conceptualized these types of violence in the Methods (lines 496-526):

3. Page 3, Line 66: The phrase “...sexual, physical, and psychological violence...” appeared in an inconsistent order. I recommend using “physical, sexual, and psychological violence” consistently throughout the manuscript, as it was more commonly used elsewhere in the text.

Thank you for catching this inconsistency, we have adjusted the phrase first in lines 46-47 and ensured the consistent use of “physical, sexual, and psychological gender-based violence” throughout the document.

4. Page 4, Line 78 to 79: The sentence “...physical, sexual, and psychological violence against women and IPV and sexual violence against women and men” should be revised to “...physical, sexual, and psychological violence against women, and IPV and sexual violence against men” for clarity.

Thank you for flagging this sentence. We have adapted the phrase in lines 88-92 to say “physical, sexual, and psychological GBV against adults (men or women aged 18 years and older)” with further explanation for how violence was defined for men and women in the Introduction (lines 94-96) Results (lines 149-152 and 228-233) and an in-depth discussion can be found in the Methods section (lines 496-526):

Lines 94-96: “We extended our analysis of GBV exposure to include physical, sexual, and psychological GBV against women by any perpetrators, and to include data on men’s exposure to physical and psychological IPV and sexual violence.”

Lines 149-152: “Importantly, due to the difficulty in identifying gendered motivations of physical violence against men, the included male-specific observations are narrowly focused on intimate partner physical violence against men to reflect this study’s emphasis on gender-based physical violence.”

Lines 228-233: “Akin to our review of physical GBV, the landscape of psychological GBV data among men is hindered by a lack of distinction between psychological violence with gender-related motivations and other forms of psychological violence. As such, the focus of our analysis of psychological GBV among male survivors concentrated on psychological IPV, which is inherently gender-related, while the focus among female survivors was much broader to align with the definitions of psychological GBV published in existing literature.”

5. Page 4, Line 91, and Page 19, Line 427:

(1) The inception date of “January 1, 1970” should be briefly justified, at least in the response letter if not in the manuscript.

Thank you for this suggestion. We started our searches in the year 1970 because this is the earliest year that many databases started their indexing. This aligns with standard practice used in other Burden of Proof studies and allows us to capture the highest quality scientific literature. We’ve included this justification in the supplementary information under Section 4.

(2) Please ensure consistency in date formatting. The manuscript used both “January 1, 1970” and “1 January 1970” (**Page 19, Line 427**).

Thank you for catching this discrepancy. We have updated the date formatting throughout the document to be consistent with January 1, 1970, including in lines 100-101 and lines 461-464.

(3) There is a discrepancy in the end dates for the studies included. Page 4 mentioned “January 31, 2023,” while Page 19 mentioned “January 31, 2021.” Additionally, the 2022 protocol published in BMJ Open specified an end date of September 30, 2021. Updated searches are common and recommended, and the authors clearly stated that an updated search was conducted in the Supplementary Information Section 4. Please correct the end date as “January 31, 2023,” where relevant throughout the manuscript.

Thank you for catching this discrepancy. While the systematic review utilized in the 2022 protocol ended in September of 2021, the systematic review was updated for this analysis to include the most recent year of data, through January 2024. This expanded search yielded 8,078 additional articles, which our team meticulously screened, leading to the inclusion of two new studies that now also inform our GBV models. This update enabled us to explore an additional association not previously covered in our original submission: the link between psychological violence and major depressive disorder. As such, we have carefully updated the figures in our results section to align with the new results based on the integration of new studies into the analysis. This process further attests to the comprehensiveness and current relevance of our work.

As suggested, we have updated the end date to be consistent throughout the document, including in lines 100-101 and lines 461-464 and ensured consistency in the supplementary information section 4 which details the dates and time frames for search updates.

6. Page 5, Line 114, Page 6, Line 130: The term “sexually transmitted infections” first appeared on Page 5, and “sexually transmitted diseases (STIs)” on **Page 9, Line 208**. Please introduce

the abbreviation “STI” here (rather than on Page 6) and use it consistently throughout the manuscript. Also, ensure consistent usage of terms like “excluding HIV/AIDS” across the text.

Thank you for highlighting this error. We have adjusted the terminology to be sexually transmitted infections excluding HIV/AIDS (STIs) throughout the document. The acronym STIs was introduced in the abstract (lines 47-51) and in the first mention of the term in the main text lines 126-129, and we have ensured consistency in use throughout the text.

7. Page 7, Line 141: Please provide the citation for “one observation which measured IPV.”

This sentence was revised based on the additional data included as a result of our expanded search through January 2024, to include 3 observations that focused on sexual IPV instead of only one as mentioned in the previous version. We added the relevant citations in lines 204-206.

8. Page 11, Line 237: The sentence “...psychological violence exposures included any psychological violence against women but for men, ...” would benefit from a comma before “but for men” to improve readability.

Thank you for correcting this error. The sentence was removed in the updated version of the manuscript.

9. Page 11, Line 241 to Line 242: Please revise “self-harm (number of included studies (n) = 3” to “self-harm (n = 3).”

The suggested adjustment has been made in line 236.

10. Page 12, Line 260 to 261: The sentence “(See Table 2 for estimated RRs and accompanying UIs, both excluding and including gamma [between-study heterogeneity], with the star rating based on the latter.)” was somewhat unclear. Please consider rephrasing for clarity.

Thank you for this suggestion. The sentence was removed in the updated version of the manuscript.

11. Page 13, Line 288 to Line 295, Line 296 to Line 306: The paragraphs effectively summarized the challenges faced by survivors and the health system’s response. However, the references (No. 94 to No. 96) in the paragraph seemed to focus more on women and abortion, which may not fully align with the broader topic discussed. Moreover, while the Brave Movement (No. 101) is significant, it is less relevant to this study which focused on adults. Additionally, barriers within the health system also impede healthcare providers from identifying and managing IPV, child abuse, and other forms of interpersonal violence. To enhance the argument and encompass a wider perspective (for example, including IPV against men), I suggest incorporating the following articles and adding one to two sentences describing the complexity of the issue at different levels:

- *Taylor JC, Bates EA, Colosi A, Creer AJ. Barriers to Men’s Help Seeking for Intimate Partner Violence. J Interpers Violence. 2022 Oct;37(19-20):NP18417-NP18444. (IPV against men)*
- *Heron RL, Eisma MC. Barriers and facilitators of disclosing domestic violence to the healthcare service: A systematic review of qualitative research. Health Soc Care*

Community. 2021 May;29(3):612-630.

(Healthcare service)

- *Li Q, Zeng J, Zhao B, Perrin N, Wenzel J, Liu F, Pang D, Liu H, Hu X, Li X, Wang Y, Davidson PM, Shi L, Campbell JC. Nurses' preparedness, opinions, barriers, and facilitators in responding to intimate partner violence: A mixed-methods study. J Nurs Scholarsh. 2024 Jan;56(1):174-190.
(Nurses' response in low-resource locations)*

Thank you for highlighting the discrepancies between references 94-96 and the corresponding discussion, and for suggesting the above sources. We have significantly revised the Discussion section to better highlight the implications of our analysis. As such, the discussion relating to the Brave Movement (no. 101) has been removed, and the overall discussion is more focused. We have removed sources 94-96 from this section and shifted focus away from discussing abortion specifically.

We reviewed all three of the recommended sources, and incorporated the Taylor JC, Bates 2022 (No. 63) and the Heron RL 2021 (No. 64) mentioned above in lines 306-309 of the Results section to discuss barriers to disclosure by men and women. Additionally, we included the Li Q 2024 (No. 74) in lines 346-349 of the Discussion to highlight systematic barriers in the health care system as suggested by the Reviewer.

12. Page 15, Line 342: Please provide the link for the website recourse (No. 107) and the website link for reference No. 3.

We have added the website link to the reference No. 4 (originally No. 3) and website recourse for reference No. 68 (originally No. 107).

13. Page 18 to Page 19, Line 423 to 425: The sentence "...a study already reported results specifically to exposure to physical or sexual intimate partner violence against women and child sexual abuse⁹" should be revised based on the following points:

(1) Citation No.9, this 2023 study published in Nature Medicine ("Health effects associated with exposure to intimate partner violence against women and childhood sexual abuse: a Burden of Proof study"), also included psychological IPV against women. Please confirm with the IHME team, and revise the sentence accordingly.

The paper referenced, Spencer et al. (2023), examines the health effects of physical and/or sexual IPV against women and childhood sexual abuse. We have confirmed with the authors that psychological IPV was not explicitly included in the reported meta-analysis in that paper. The only way psychological IPV was considered in Spencer et al. (2023) was as a component of accepted alternate case definitions that also included physical and/or sexual IPV. No data points only focused on psychological IPV were included in the Spencer meta-analysis nor was psychological IPV discussed in the paper. In comparison, our present manuscript does include a handful of data points that focus solely on psychological IPV in our psychological violence models, but it does not include combined exposure definitions that span multiple forms of GBV (or IPV), unlike many of those included in Spencer et al. (2023).

(2) The 2023 Nature Medicine study included 57 articles on IPV against women, some of which were also included in the current study (eg, Leung 2002 from Hong Kong, Ibrahim 2015 from

Egypt). Is there any overlap in the findings between this article and the current study? If so, what is the proportion of overlap? If not, please clarify this briefly in the manuscript.

There is no direct overlap in the findings between Spencer et al. (2023) because, while some of our underlying studies have been used in both papers, the scope of our analysis is quite different. Namely, our analysis parses out the distinct health effects of physical, sexual, and psychological GBV without distinguishing the gender of the survivors. The Spencer et al. (2023) analysis looks specifically at physical and/or sexual IPV as a composite exposure category among women only. Our analyses include some of the same data points where a study reported either an effect size for physical IPV or an effect size for sexual IPV, but it does not include composite exposure categories. Furthermore, our analyses also include observations on non-partner GBV and men, and the data point selection process we used was different – when some studies may have reported effect sizes for composite physical and/or sexual IPV as well as exposure-stratified effect sizes, we would have selected the exposure-specific results while the Spencer et al. (2023) analysis would have selected the composite result. Similarly, some studies may have reported an effect size for violence among men and women as well as sex-stratified values. We would have selected the combined-sex effect size, while Spencer et al. (2023) would have only used the women-specific value. Each option better matches the case definitions used by the respective research teams.

We have clarified some of these differences in Methods (lines 508-526) and we also added a new section in the Supplementary Information comparing the two papers (Supplementary Information Section 5):

Lines 508-526 “Furthermore, only exposure to these forms of GBV at or above the age of 18 was investigated. Results examining the health effects of childhood physical abuse, psychological abuse, and neglect using the aforementioned systematic review are reported in a concurrent study¹¹⁹. A previously published analysis examined the health consequences of childhood sexual abuse, as well as exposure to physical and/or sexual intimate partner violence against women¹⁴ (Supplementary Table S15 and Figure S23). While our analyses do not focus on violence perpetrated by partners, physical IPV and sexual IPV are considered sub-groups of physical GBV and sexual violence, respectively. However, our study complements this prior analysis by 1) including more recent literature, 2) examining psychological GBV, 3) narrowing in on adulthood exposures of GBV rather than including the multi-faceted period of adolescence, 4) parsing out the distinct effects of physical GBV and sexual violence rather than their combined exposure, and 5) incorporating both partner-perpetrated GBV and violent acts perpetrated by non-partners because of an individual’s sex, gender identity, gender expression, or expression of masculinity and femininity. Furthermore, our analysis is not limited by the gender of the survivor of violence and includes GBV perpetrated against men. However, due to a paucity of data distinguishing the motivations of physical or psychological violence perpetrated against men due to their sex or gender, rather than perpetrated against men due to other factors, the scope of the studies on physical or psychological GBV against men, specifically, is largely limited to physical or psychological IPV. In comparison, the studies capturing physical or psychological GBV against women that were included in the present analysis have a much broader range of perpetrators.”

(3) How does the current study supplement the 2023 Nature Medicine study and the Lancet Commission on Gender-based Violence regarding the included types of violence against women? Given the differing terms and definitions used in IPV research, any vague expressions

could lead to confusion. Please provide an additional sentence clarifying these points, possibly on **Page 19, Line 425**.

Importantly, this work builds off the 2023 study to quantify and evaluate the health effects of GBV by taking a broader lens on GBV exposure as a health risk among adults. While we continue to include all forms of IPV against women in this research, we also extended our analysis of GBV exposure to include physical, sexual, and psychological GBV against women by non-partner perpetrators, and include data that characterized men’s exposure to physical and psychological IPV and sexual violence. The 2023 estimates of the health risks associated with physical and/or sexual IPV against women, specifically, have been reported separately and include data up to January 31, 2023, while the current estimates present an expanded view – with one more additional year of data – of health risks for physical, sexual, or psychological GBV, distinctly, against any adult victim by any perpetrator. Ultimately, these two papers complement each other to provide a comprehensive view of the health effects of gender-based violence across its various dimensions. Both sets of analyses are being used to inform the upcoming report on the Lancet Commission on Gender-based Violence.

We have included updated language in lines 88-99 to better articulate this information in the updated manuscript:

Lines 88-99: “Our study—which contributes to the goals of the Lancet Commission on Gender-based Violence and Maltreatment of Young People² to expand the consideration of GBV as an important global health concern for all people—provides a comprehensive systematic review of the health effects associated with exposure to physical, sexual, and psychological GBV against adults (men and women aged 18 years and older). This work builds off a prior effort to quantify and evaluate the health effects of IPV and Childhood Sexual Abuse^{12,13} by taking a broader lens on GBV exposure and its associated health consequences. We extended our analysis of GBV exposure to include physical, sexual, and psychological GBV against women by any perpetrators, and to include data on men’s exposure to physical and psychological IPV and sexual violence. Expanding our exposure scope allows us to disentangle the different manifestations of GBV to provide a more accurate representation of the complex relationships between GBV exposure and health outcomes beyond the common focus of IPV among women and, in the future, to allow for a more comprehensive estimation of the disease burden attributable to GBV.”

14. Page 19, Line 437: Please add the closing parenthesis “)” to the sentence.

Thank you for catching this grammatical error. The sentence was removed in the updated manuscript.

15. Page 19, Line 442: Please delete the word “is” in the phrase “including but is not limited to.”

The referenced sentence has been revised to the following in lines 503-505: “Sexual violence, considered inherently a form of GBV, is any deliberate, unwanted, and non-essential sexual act against an adult, including both completed and attempted rape, sexual assault, and non-contact sexual acts.”

16. Page 20, Line 469: The term “gender-based violence” was first mentioned on **Page 4 Line 80**. Please introduce the abbreviation “GBV” on Page 4 and use it consistently throughout the manuscript. Check for consistency with other terms as well.

Thank you for highlighting these discrepancies. We have introduced the acronym GBV where gender-based violence is first introduced in the manuscript (lines 46-47 of the abstract and lines 62-63 in the main text) ensured consistency throughout the document. We have also done a thorough review of the document to ensure the consistency of terminology utilized across the document.

17. Page 22, Line 507: Please correct the word “selected” by using the past tense.

Thank you for catching this grammatical error. We adjusted the word “select” in lines 554-556 to reflect the past tense.

18. Page 22, Line 511: The manuscript contained inconsistent use of hyphen (-), shorter en dash (–), and longer em dash (—). For example, in “Supplementary Tables S18–S19,” the shorter en dash should be used, not the hyphen (-). Please correct these issues throughout the manuscript and supplementary materials. Although time-consuming, these revisions are essential, especially for an article from the IHME.

Thank you for catching these grammatical errors. We have reviewed the entire document as well as the supplementary materials and revised our use of hyphens, en dashes and em dashes as suggested.

19. Page 26, Line 591: Please insert a space between “In” and “(BPRF).”

Thank you for flagging this error. The sentence was removed in the updated manuscript.

20. Supplementary information Figure S1: Please re-create the screenshot of the PRISMA diagram to remove unnecessary error notifications, such as those under “screened,” “excluded,” and “retrieval.”

We appreciate the reviewer noting this oversight. We have thoroughly updated our PRISMA diagram given the update to our systematic review, and we have confirmed that the updated screenshot does not have any unnecessary error notifications.

Reviewer #3:

1. This study estimates the health-related impacts of exposure to multiple forms of violence in adulthood. It expands upon existing efforts to measure the health impacts of exposure to violence against women by their male intimate partners by including physical, sexual, or psychological violence perpetrated by anyone against adult women and IPV or sexual violence against adult men. As such it makes an important, new contribution to the on-going effort to establish exposure to violence as a significant risk factor contributing to the global burden of disease. The findings should be of immediate interest to the fields of public health, victimology, criminology, social work, as well as those particularly interested in violence against women and other forms of gender-based abuse.

We appreciate the reviewer’s recognition of our findings’ relevance to an array of related fields, and of how our research advances awareness of exposure to various forms of GBV as an important and preventable health risk factor.

2. The scope of the review, however, could be better justified and more consistently expressed. **The abstract** initially claims, for example, to examine the health impacts of violence—which

could suggest that among the exposures examined would be things like community or gang violence and simple assault. The abstract goes on to clarify that the focus is on violence by all perpetrators against women, and IPV or sexual violence against men. Later in the Introduction the authors describe their work as: “the most extensive systematic review on the health effects associated with exposure to violence.” Similar slippages to “health effects of violence” happen throughout the manuscript.

We have substantially revised the language in both the abstract (lines 46-51) and introduction (lines 88-96) and the throughout the manuscript to clarify our focus on the health effects of various forms of GBV, rather than violence in its entirety. Additionally, we have also revised sections of the Discussion (lines 149-152 and 228-233) to clarify the rationale for our focus and updated the language in the Methods (lines 496-526) for additional clarity. We believe that the updated language better communicates the focus on the forms of violence of interest:

Lines 46-51: “The health impacts of exposure to physical, sexual, or psychological gender-based violence (GBV) against men and women during adulthood are substantial yet not well delineated. Our comprehensive systematic review and meta-analysis evaluated the associations between GBV (including but not limited to intimate partner violence) and eight health outcomes: sexually transmitted infections (STIs) excluding HIV, maternal abortion and miscarriage, HIV/AIDS, major depressive disorder, anxiety disorders, drug use disorders, alcohol use disorders, and self-harm.”

Lines 88-96: “Our study—which contributes to the goals of the Lancet Commission on Gender-based Violence and Maltreatment of Young People² to expand the consideration of GBV as an important global health concern for all people—provides a comprehensive systematic review of the health effects associated with exposure to physical, sexual, and psychological GBV against adults (men and women aged 18 years and older). This work builds off a prior effort to quantify and evaluate the health effects of IPV and Childhood Sexual Abuse^{12,13} by taking a broader lens on GBV exposure and its associated health consequences. We extended our analysis of GBV exposure to include physical, sexual, and psychological GBV against women by any perpetrators, and to include data on men’s exposure to physical and psychological IPV and sexual violence.”

3. Moreover, the authors do not offer a clear explanation for the types of violence they have chosen to include. One could ask why include physical violence against women by acquaintances or strangers but not for men? Nor does the search strategy suggest that the types of physical assault that might be captured in crime data have been consciously included for either men or women. This leads me to believe that the goal is not to establish the full health effects of victimization as a risk factor for health outcomes, but to update the existing risk factor analysis on violence against women by adding data on IPV and sexual assault against men. In another place, the analysis is framed as analysis of the health impact of violence directed at specific groups of adults, presumably because of their group identity. If this were the underlying rationale, one would expect search terms for hate crimes and physical and sexual violence against sexual and gender minorities to be included.

Presumably, the framing of the review (and hence types of studies included), is either a function of the fact that the analysis is derived from a larger study on violence against women or because the review is intended to focus on gender-based violence only—that is types of violence that are influenced, at least in part, by gender-related norms and power relations. It

would be useful for the authors to describe what led them to their decisions on which types of violence to include (in terms of the author's policy intent, paradigm, or circumstance, e.g., membership on the Lancet Commission on GBV and Maltreatment of Children).

The reviewer is correct that the focus of the review was GBV (in the reviewer's words, "types of violence that are influenced, at least in part, by gender-related norms and power relations," and we appreciate the reviewer pointing out the ambiguities in the manuscript around this point.

We clarified the language throughout the manuscript to specify that our analysis examines the health effects of three specific forms of GBV, rather than any form of victimization. We have similarly revised our Introduction (lines 88-96) and Discussion (lines 149-152 and 228-233) as noted above, with a particular focus on why we structured our analysis around the different types of GBV and how our choices inform the interpretation of our findings. We updated the language in the Methods (lines 496-526) regarding how we defined our exposures of interest selected our exposures and the corresponding rationale, including the focus on physical and psychological IPV for men:

Lines 496-507: "While the systematic review captured all forms of GBV or VAC, the current analysis focused on distinct forms of GBV experienced during adulthood: physical GBV, psychological GBV, and sexual violence. For the purposes of the present analysis, we defined physical GBV as deliberate, unwanted, and non-essential act of physical force against the body of an adult due to aspects of their identity related to gender. Similarly, psychological GBV was defined as deliberate, unwanted, and non-essential verbal or non-verbal acts driven by gender-related components of the victim's gender that result in long-term psychological harm. These acts can include terrorizing, harassing, spurning, humiliating, and controlling. Sexual violence, considered inherently a form of GBV, is any deliberate, unwanted, and non-essential sexual act against an adult, including both completed and attempted rape, sexual assault, and non-contact sexual acts. The definitions operationalized here are based on and adapted from similar categories proposed in the International Classification of Violence against Children¹⁰¹ but have a narrow focus on specific forms of GBV.

Data & methodology:

4. The manuscript is a systematic review and meta-analysis of existing literature that assesses the potential incremental risks posed by exposure to physical, sexual, or psychological violence in adulthood. The methods for the overarching systematic review and meta-analysis are well documented and well-executed.

Thank you to the reviewer for their praise of our systematic review and meta-analytic methods. We appreciate the recognition of this work.

5. I am unable to evaluate the underlying Burden of Proof Risk Function methodology, because the methods exceed my quantitative skills. I do have several suggestions and questions, however, regarding other elements of the methodology.

We appreciate the reviewer's feedback regarding the methodology and have provided in-depth responses to each comment below. Furthermore, we have clarified some of the language around the Burden of Proof methodology to facilitate understanding (lines 100-113, 425-431, and 637-659):

Furthermore, we clarified the strengths of the Burden of Proof methodology throughout the manuscript in response to reviewer comments, emphasizing how the methodology allows us to synthesize the available evidence to more systematically generate estimates of association and to formally assess evidence strength:

Lines 100-113: “Drawing from a comprehensive systematic review of the literature published between January 1, 1970 and January 31, 2024, we estimated the associations between physical, sexual, and psychological GBV against men and women and a range of health outcomes using the Burden of Proof methodology developed by Zheng et al¹⁴. This methodology allows us to both systematically evaluate the potential association between the exposure of interest and a given health outcome and to quantify the strength of the underlying evidence. In addition to producing conventional measures of association, the Burden of Proof methodology generates conservative measures that account for both known and unknown sources of heterogeneity across input studies. The Burden of Proof Risk Function (BPRF) can be translated into both a Risk-Outcome Score (ROS) and an estimate of the minimum percent of increased health risk attributable to GBV exposure. The ROS communicates both the magnitude of the association and the strength of the underlying data, with greater positive values reflecting a larger effect size and/or stronger evidence. The ROS can in turn be converted into easily comparable and interpretable star ratings ranging from one (weak) to five (strong) that categorize significant associations according to effect size and evidence strength.”

Lines 425-431: “In this study, the Burden of Proof (BoP) methodology was applied to estimate the association between exposure to gender-based physical, sexual, and psychological violence during adulthood (modeled as dichotomous risk factors) and selected health outcomes and to evaluate the strength of evidence underlying the estimates of association. We applied the BoP methodology if a risk-outcome pair had at least three studies identified in the scientific literature. We generated estimates of relative risks, burden of proof risk function (BPRF), and risk-outcome score (ROS) for all risk-outcome pairs in a single model with no location- or age-specific results.”

Selection of Studies:

6. As I was reading and checking references, I repeatedly thought of studies that I would have expected to see included that were not. For example:

- *Kate Doyle, Ruti G. Levto, Emmanuel Karamage, et al. Long-term impacts of the Bandebereho programme on violence against women and children, maternal health seeking, and couple relations in Rwanda: a six-year follow-up of a randomised controlled trial. eClinicalMedicine 2023;64: 102233 Published Online 26 September 2023 <https://doi.org/10.1016/j.eclinm.2023.102233>*
- *Chatterji, S., Heise, L., 2021. Examining the bi-directional relationship between intimate partner violence and depression: findings from a longitudinal study among women and men in rural Rwanda. SSM-Mental Health 1, 100038.*
- *T. Muhammad, Saddaf Naaz Akhtar, Waad Ali, Chanda Maurya, Cross-lagged relationships between exposure to intimate partner violence, depressive symptoms and suicidal thoughts among adolescent and young married women, Journal of Affective Disorders, Volume 360,2024,Pages 259-267,ISSN 0165-0327,<https://doi.org/10.1016/j.jad.2024.05.088>.*
- *Devries, K.M., Mak, J.Y., Bacchus, L.J., et al., 2013. Intimate partner violence and incident depressive symptoms and suicide attempts: a systematic review of longitudinal studies. PLoS Med. 10 <https://doi.org/10.1371/journal.pmed.1001439>.*

Part of the problem is that the discussion of methods comes at the end of the article, which leaves the reader wondering the boundaries of inclusion. Even in the delayed methods section, nowhere does it clearly state the inclusion and exclusion criteria for studies. It wasn't until I examined the supplementary material that I came to realize what types of studies (cohort, case control, etc) and what type of data (administrative, survey, police reports, etc.) qualified for establishing a risk association. Also, very important from the perspective of the violence against women field, is the decision to exclude studies that use combined outcome measures like physical and/or sexual violence by an intimate partner. I would suggest moving a combined version of Tables S13 and S14 to the methods section or at least clarifying early in the manuscript the types of studies and outcomes that qualify for inclusion in the review.

Although limits on word count and number of tables and figures restrict our ability to move Tables S13-14 to the main text of the manuscript, we have added detail to descriptions of our inclusion and exclusion criteria in the Methods (lines 478-486):

Lines 478-486: "To merit inclusion, studies needed to 1) use an eligible study design (cohort, case-control, or case-crossover) that allowed the research team to determine temporality between the violence exposure and development of the health outcome, 2) report a measure of association or enough detail to derive a measure of association between GBV/VAC exposure and a health outcome, and 3) appropriately define the exposure and outcomes. For example, composite measures of violence including non-gender-based violence were not eligible for inclusion. Studies were excluded if they 1) used cross-sectional, ecological, case series, or case study designs, 2) failed to establish temporality, or 3) reported incomplete data. More details on the inclusion/exclusion criteria can be found elsewhere^{12,14} and in the Supplementary Tables S12 and S13."

With specific regard to the four papers highlighted by the reviewer, we checked our systematic review to confirm whether or not they had been captured. For those that had been captured, we re-screened them to confirm their exclusion or to retroactively include them.

- Doyle et al. (2023) was captured in our systematic review and screened for potential inclusion. However, it was excluded because of the outcome definitions used. The study's potentially usable data evaluated the association between violence and depressive symptoms, which – although related to depressive disorders – did not match the diagnostic requirements used to define major depressive disorders or dysthymia in the Global Burden of Disease Study. As a result, we have affirmed that the study was correctly excluded. The details regarding the diagnostic tools accepted can be found in our Supplementary Information Table S18.
- Chatterji et al. (2021) was similarly captured and screened in our systematic review. It also examined depression, but, unlike Doyle et al., the Chatterji study used the Centre for Epidemiologic Studies Depression self-report measure (CESD-10), a diagnostic measure that falls under our included tools. However, the study used a cutoff value of 10 for this measure rather than our accepted standard of 16. As a result, it was also excluded due to having an outcome definition that did not meet our inclusion criteria.
- The publication date of May 2024 for the paper by Muhammad et al. falls outside of our search dates, occurring after both our original submission of this manuscript and the end of our updated review period. It will be captured in future updates to our systematic review.

- Devries et al., as a systematic review of longitudinal studies, did not meet our inclusion criteria but was reviewed in our original systematic review for its underlying studies. Any potentially relevant underlying studies were screened as part of our review and, if eligible, included in our final models.

Outcome inclusion.

7. With respect to IPV, I find the decision to exclude studies with a composite outcome measure from the review, problematic. Historically, the standard measure for studies on IPV has been physical and/or sexual violence by a current and/or former partner. This is because most women experience IPV as a pattern of different types of violence and controlling behaviors over time. When creating a “case” of violence in the context of prevalence or risk factor studies, the traditional physical and/or sexual violence measure captured the mix experiences that women report who meet the definition of IPV. Most of those who experience sexual IPV or physical IPV also experience emotional abuse and episodes that include amalgams of different types. I can see the benefits of exploring the different types of violence separately, but I suspect that if you additionally had allowed for composite measures for IPV, you would have had more studies qualifying for inclusion.

We agree with the reviewer’s concern regarding composite measures of IPV, which is why a previous publication from the same systematic review (Spencer et al., 2023) explicitly examined the health effects of physical and/or sexual IPV among women. The prior publication uses the standard measure provided by the reviewer as well as an expanded age cut-off of 15 years old to capture adolescence when many individuals may experience IPV for the first time. The scope and aim of the present analysis were very different from the prior publication focused solely on IPV. Namely, we are focused on exploring the distinct health consequences of physical, psychological, and sexual GBV (including, but not limited to, IPV) as separate exposure categories for adults.

We acknowledge and discuss (lines 367-386) that our approach of parsing out these three forms of GBV likely under-estimates the health risks experienced by a survivor of GBV given the frequent co-occurrence of these forms of violence, but they reflect the minimum additional risk an individual may experience as a survivor of at least the specific form of GBV in question:

Lines 367-386: “Similarly, we limited our existing analyses to studies that reported distinct effect sizes for physical, sexual, and/or psychological GBV to parse out the unique health consequences of each of these types of GBV. However, most studies did not explicitly state whether the exposure definition they used was restricted to individuals who had experienced only the GBV type of interest, or if it potentially included individuals who had experienced other types of GBV as well. In the latter case, the resulting effect size may overestimate the distinct health effect of the single GBV type, particularly since individuals who experience one form of violence are more likely to experience other forms^{109,110}. Studies also very rarely clarified whether the reference group of unexposed individuals excluded or included people who had experienced other forms of GBV, which—conversely—may result in underestimating effect sizes. In the absence of more detailed reporting on the part of included studies, any adjustment or bias covariate to account for other forms of violence would require substantial assumptions on the part of the research team, which may, in turn, introduce additional sources of uncertainty to the model. When input studies provided information regarding co-occurrence with other forms of violence, we prioritized data points that were limited to groups exposed only to the GBV-type of

interest and reference groups with no other violence exposure. We also prioritized observations that controlled for any other form of violence to reduce the interference of ill-defined exposure groups in order to focus specifically on the health consequences of physical, sexual, or psychological GBV. However, in the interpretation of our findings, it is important to consider that different types of GBV rarely occur independently. As such, our estimates likely represent the lower bound of risk experienced by an individual exposed to any of these forms of GBV with co-occurrence leading to compounding health consequences.

In our Methods section, we also made revisions to clarify our exposure definitions of interest, laid out how this affected our data selection, and drew a direct comparison to the exposure used in Spencer et al. (2023) focused on IPV (lines 496-526, Supplementary Table S15 and Figure S23):

Lines 512-521: “While our analyses do not focus on violence perpetrated by partners, physical IPV and sexual IPV are considered sub-groups of physical GBV and sexual violence, respectively. However, our study complements this prior analysis by 1) including more recent literature, 2) examining psychological GBV, 3) narrowing in on adulthood exposures of GBV rather than including the multi-faceted period of adolescence, 4) parsing out the distinct effects of physical GBV and sexual violence rather than their combined exposure, and 5) incorporating both partner-perpetrated GBV and violent acts perpetrated by non-partners because of an individual’s sex, gender identity, gender expression, or expression of masculinity and femininity. Furthermore, our analysis is not limited by the gender of the survivor of violence and includes GBV perpetrated against men.”

8. I also think you would have a wider evidence base if you had not restricted studies to only women over 18; there are a number of well-known prospective cohorts of young people being followed (e.g. Ecuador, Peru, Vietnam, India) overtime that collect information on IPV and health outcomes. Many of these follow adolescents into their adult years (such as the Young Lives Study out the University of Oxford or the UDAYA study cited above).

We would like to clarify that we did not restrict studies to only women over 18 in the overarching systematic review. Rather, in the review, we had no restrictions on gender or age of the participants for included studies. In doing so, we captured studies like those mentioned by the reviewer – prospective cohort studies spanning several age periods. Other studies using data from the same systematic review, including Spencer et al. (2023) mentioned above and a publication currently in preparation by this research team, focus on exposure to forms of violence at other ages. Spencer et al. (2023) narrowed in on data pertaining to the health effects of physical and/or sexual IPV experienced at or above the age of 15, while the publication in preparation focuses on violence against children defined below the age of 18. In comparison, the scope of our analysis is to examine the health effects of physical, psychological and sexual GBV experienced during adulthood. As such, we subset the data extracted during the overarching systematic review to only observations that define exposure as instances occurring during adulthood. This may include long-term cohort studies that otherwise meet our inclusion criteria so long as the study provides data on the health consequences of physical, psychological, or sexual GBV among individuals who were exposed after they turn 18.

We have added a point in the Discussion section (lines 399-410) highlighting how this dichotomy restricts the interpretability of our GBV findings because it does not capture the

period of adolescence, however, we believe that this restriction allows us to narrow in on the health consequences of adulthood exposure, specifically:

Lines 399-410: “Lastly, our analysis focuses on exposures to GBV that occur during adulthood, defined as at or above the age of 18. A concurrent analysis based on the same systematic review is examining the health effects of physical abuse, psychological abuse, and neglect against children below the age of 18, and a previously published Burden of Proof analysis using prior iterations of the systematic review focused on childhood sexual abuse and on physical and/or sexual IPV above the age of 15¹³. In totality, these works present a comprehensive view of the health consequences of GBV and VAC across the lifespan. However, it is important to highlight the role that GBV experienced during adolescence, particularly non-partner perpetrated forms of GBV, may play in the overall health toll of GBV. This transitional adolescent period is not reflected in the present analysis despite the prevalence of GBV during these ages. We focused on forms of violence that did not overlap with the categories of violence considered to be VAC due, in part, to the differences in potential health interventions, but GBV during adolescence merits further focused investigation as an impactful risk factor for future health.”

9. Additionally, it seems that you may be missing relevant data on IPV and health outcomes from prospective cohorts embedded in impact evaluations/RCTs. Both the Doyle and Chatterji analysis come from evaluation of IPV prevention interventions in Rwanda.

We are always looking to add more data to our models given that new studies may both bolster our existing analyses and add new potential risk-outcome associations that are feasible to examine. As such, we were concerned by the possibility that we may be systematically missing relevant data. We returned to the original systematic review and identified all excluded studies captured by the search strings that contained the words “randomized control trial” or “RCT” in the title or abstract. We identified 1149 such studies and re-screened them. Unfortunately, we did not find any data that was usable for our analyses. We have included a table of the reasons for their exclusions below:

Exclusion Reason	Number of Sources Excluded
Irrelevant on the basis of Title/Abstract	625
Wrong Outcome Outcome is not included in our study or outcome does not meet GBD definition	363
Wrong Exposure Exposure is a violence type not included in our study, or source only includes data on cumulative exposure types including one or more exposure type which does not meet our inclusion criteria	84
Population Study population is non-representative, or study does not include a non-exposed group	61
Temporality Study provides cross sectional data only: temporality cannot be established	16

10. Obviously, it is the authors prerogative to establish the inclusion and exclusion criteria, but I suspect some of the decisions made have needlessly reduced the data available for inclusion in the systematic review.

We believe our new language, including the addition of a section on exposure definitions (lines 496-526) that addresses the two areas highlighted by the reviewer, may alleviate the concern

regarding constraints on the dataset. Our inclusion and exclusion criteria for the systematic review were established prospectively and in accordance with systematic review best practices and follow established Global Burden of Disease Study case definitions, while the definitions used for the current analysis are consistent with the scope and aims of our meta-analysis.

Lines 496-507: “While the systematic review captured all forms of GBV or VAC, the current analysis focused on distinct forms of GBV experienced during adulthood: physical GBV, psychological GBV, and sexual violence. For the purposes of the present analysis, we defined physical GBV as deliberate, unwanted, and non-essential act of physical force against the body of an adult due to aspects of their identity related to gender. Similarly, psychological GBV was defined as deliberate, unwanted, and non-essential verbal or non-verbal acts driven by gender-related components of the victim’s gender that result in long-term psychological harm. These acts can include terrorizing, harassing, spurning, humiliating, and controlling. Sexual violence, considered inherently a form of GBV, is any deliberate, unwanted, and non-essential sexual act against an adult, including both completed and attempted rape, sexual assault, and non-contact sexual acts. The definitions operationalized here are based on and adapted from similar categories proposed in the International Classification of Violence against Children⁷⁰ but have a narrow focus on specific forms of GBV.”

We have also clarified our language related to inclusion and exclusion of studies throughout the Methods section, particularly regarding our exposure definition and systematic review protocol (lines 478-486):

Lines 478-486: “To merit inclusion, studies needed to 1) use an eligible study design (cohort, case-control, or case-crossover) that allowed the research team to determine temporality between the violence exposure and development of the health outcome, 2) report a measure of association or enough detail to derive a measure of association between GBV/VAC exposure and a health outcome, and 3) appropriately define the exposure and outcomes. For example, composite measures of violence including non-gender-based violence were not eligible for inclusion. Studies were excluded if they 1) used cross-sectional, ecological, case series, or case study designs, 2) failed to establish temporality, or 3) reported incomplete data. More details on the inclusion/exclusion criteria can be found elsewhere^{12,14} and in the Supplementary Tables S12 and S13.”

Outcome construction.

11. One additional thing I did not see addressed is whether or not different investigators used a “clean” reference group when constructing their IPV exposure variable. Since individuals experiencing IPV so often experience multiple forms of violence during any episode or over time, binary variables based on “have experienced” versus “have not experienced” tend to underestimate the strength of association unless a special effort is made to remove women who have experienced any other type of IPV from the reference group. If women experienced a combination of physical and sexual violence from the same partner and she is coded “yes” = 1 for physical violence, and everyone else becomes zero, then women experiencing other types of IPV will be in the reference group, thus “diluting” the strength of association between physical violence and IPV. There is a mix of practice in the field regarding “clean” versus “non-clean” reference groups, so you may want to add this as a bias test in your meta-analysis.

This is discussed in the Devries et al and Chatterji et al articles above and in this measurement brief:

As the reviewer points out, there is a large degree of variation in the field regarding the use of “clean” exposure definitions, and we found this to be true within the studies identified in our analysis as well. We extracted whether or not an analytical sample was restricted to individuals who had either only experienced a violence type of interest or who had never experienced any form of violence, but we unfortunately found that we were unable to use this data given the degree of ambiguity in the way studies tend to report their reference groups. Any use of this information to be incorporated in our analysis would require multiple assumptions by the research team, which could be flawed and introduce bias into the analysis. For the studies where the definition of the reference group was made explicit and where there were several eligible observations for the health outcome of interest, we made sure to select the observations with restricted reference groups or where other forms of GBV exposure were controlled for.

We have clarified our approach to determining the case definitions of our exposures of interest in the Methods (lines 496-507):

Lines 496-507: “While the systematic review captured all forms of GBV or VAC, the current analysis focused on distinct forms of GBV experienced during adulthood: physical GBV, psychological GBV, and sexual violence. For the purposes of the present analysis, we defined physical GBV as deliberate, unwanted, and non-essential act of physical force against the body of an adult due to aspects of their identity related to gender. Similarly, psychological GBV was defined as deliberate, unwanted, and non-essential verbal or non-verbal acts driven by gender-related components of the victim’s gender that result in long-term psychological harm. These acts can include terrorizing, harassing, spurning, humiliating, and controlling. Sexual violence, considered inherently a form of GBV, is any deliberate, unwanted, and non-essential sexual act against an adult, including both completed and attempted rape, sexual assault, and non-contact sexual acts. The definitions operationalized here are based on and adapted from similar categories proposed in the International Classification of Violence against Children⁷⁰ but have a narrow focus on specific forms of GBV.”

Furthermore, we added language highlighting the issue of clean reference groups in the Limitations (lines 367-386), as we concur with the reviewer that this challenge may result in attenuated effects:

Lines 367-386: “Similarly, we limited our existing analyses to studies that reported distinct effect sizes for physical, sexual, and/or psychological GBV to parse out the unique health consequences of each of these types of GBV. However, most studies did not explicitly state whether the exposure definition they used was restricted to individuals who had experienced only the GBV type of interest, or if it potentially included individuals who had experienced other types of GBV as well. In the latter case, the resulting effect size may overestimate the distinct health effect of the single GBV type, particularly since individuals who experience one form of violence are more likely to experience other forms^{76,77}. Studies also very rarely clarified whether the reference group of unexposed individuals excluded or included people who had experienced other forms of GBV, which—conversely—may result in underestimating effect sizes. In the absence of more detailed reporting on the part of included studies, any adjustment or bias covariate to account for other forms of violence would require substantial assumptions on the part of the research team, which may, in turn, introduce additional sources of uncertainty to the model. When input studies provided information regarding co-occurrence with other forms of

violence, we prioritized data points that were limited to groups exposed only to the GBV-type of interest and reference groups with no other violence exposure. We also prioritized observations that controlled for any other form of violence to reduce the interference of ill-defined exposure groups in order to focus specifically on the health consequences of physical, sexual, or psychological GBV. However, in the interpretation of our findings, it is important to consider that different types of GBV rarely occur independently. As such, our estimates likely represent the lower bound of risk experienced by an individual exposed to any of these forms of GBV with co-occurrence leading to compounding health consequences.”

12. Overall, the manuscript is an important contribution to the literature. It would make sense in the future to do a similar analysis of the impact of any victimization in childhood, adolescence and adulthood on negative health outcomes and to explore if the risk of negative outcomes increases as the number and types of violence experiences likewise increase.

In parallel to this research, our team has also evaluated existing evidence on the health effects of childhood maltreatment, including during adolescence, as we agree with the reviewer that this is an important complement to our present analysis. Moving forward, we hope to continue extending our analysis across ages and considering frequency and co-occurrence of violence, particularly as the evidence base grows to make such extensions feasible.

Institute for Health Metrics and Evaluation

Population Health Building / Hans Rosling Center
3980 15th Ave. NE, Seattle, WA 98195 USA@
UW Campus Box #351615

28 January 2025

Dr. [REDACTED]
[REDACTED]

Nature Human Behaviour

Resubmission: The health effects associated with physical, sexual and psychological gender-based violence against men and women: a Burden of Proof study - NATHUMBEHAV-24020709

Dear Dr. [REDACTED]

I am writing on behalf of my colleagues and myself to express our pleasure in submitting the revised version of our manuscript titled "*The Health Effects Associated with Physical, Sexual, and Psychological Gender-Based Violence Against Men and Women: A Burden of Proof Study*" for consideration for publication in Nature Human Behaviour. We are grateful for the opportunity to address the feedback provided in the Author Checklist. To ensure thoroughness, we have meticulously responded to each point raised, making the necessary revisions to enhance our manuscript.

We wish to extend our sincere thanks to both the reviewers and editors for their invaluable final insights concerning our manuscript. In response to the suggestions provided in the Author Checklist, we have incorporated all recommended changes into the Main Text and Supplementary Information files. This includes adopting the suggested title, relocating the PRISMA diagram from the Supplementary Information to the Main Text, renumbering all figures across both files, conducting a thorough review of all meta-analyzed studies mentioned within the manuscript, and applying all advised formatting modifications to our Figures and Tables. In this updated version, we have also slightly adapted the language surrounding our definition of exposure to focus on gender-based violence as the exposure and omit mentions of adulthood exposure. This is in part motivated by the fact that the definition of adulthood as starting above the age of 18 is not fully consistent with measuring gender-based violence, which frequently occurs among teenagers. This change in language also better aligns with the most common exposure period (lifetime recall) observed in the studies that underlie our analysis. Given the lack of a clear age distinction in these studies, we believe it is more accurate to refer to GBV without incorporating an age-related marker of exposure. This adjustment is linguistic and does not alter any of our

results, conclusions, or discussion points but more accurately reflects the available evidence.

Additionally, we are submitting the finalized versions of the Editorial Policy Checklist, Reporting Summary, and Inventory of Supporting Information. We have also collected all Conflict-of-Interest forms from the contributing authors of the manuscript.

The data associated with our study will be accessible on the GHDx website (<https://ghdx.healthdata.org/record/ihme-data/gbv-health-effects-bop-risk-outcome-scores>) upon publication of the article. Additionally, all relevant data are currently available in the Supplementary Information file for immediate reference.

We consent to the publication of reviewer comments, author rebuttal letters, and editorial decision letters as Supplementary material.

Please let us know if we can provide any further information.

Best Regards,

Emmanuela Gakidou, PhD

Senior Director of Organizational Development and Training
Professor of Health Metrics Sciences
Institute for Health Metrics and Evaluation
University of Washington, 2310 5th Ave Suite 600
Seattle, WA 98121, USA
Email: gakidou@uw.edu